# WEIGHTED CONDITIONAL FLOW MATCHING

## ABSTRACT

Conditional flow matching (CFM) has emerged as a powerful framework for training continuous normalizing flows due to its computational efficiency and effectiveness. However, standard CFM often produces paths that deviate significantly from straight-line interpolations between prior and target distributions, making generation slower and less accurate due to the need for fine discretization at inference. Recent methods enhance CFM performance by inducing shorter and straighter trajectories but typically rely on computationally expensive mini-batch optimal transport (OT). Drawing insights from entropic optimal transport (EOT), we propose *weighted conditional flow matching* (W-CFM), a novel approach that modifies the classical CFM loss by weighting each training pair $(x, y)$ with a Gibbs kernel. We show that this weighting recovers the entropic OT coupling up to some bias in the marginals, and we provide conditions under which the marginals remain nearly unchanged. Moreover, we establish an equivalence between W-CFM and the minibatch OT method in the large-batch limit, showing how our method overcomes computational and performance bottlenecks linked to batch size. Empirically, we test our method on unconditional generation on various synthetic and real datasets, confirming that W-CFM achieves comparable or superior sample quality, fidelity, and diversity to other alternative baselines while maintaining the computational efficiency of vanilla CFM.

## 1 INTRODUCTION

Generative modeling seeks to learn a parameterized transformation that maps a simple prior (e.g., a Gaussian) to a complex data distribution. Continuous normalizing flows (CNFs) instead train a time-dependent vector field to solve an ordinary differential equation (ODE) that transports base samples to data samples with exact likelihood computation and invertibility. However, training CNFs by likelihood maximization suffers from training instability and fails to scale efficiently to large or high-dimensional datasets (Chen et al., 2018; Grathwohl et al., 2018; Onken et al., 2021). Flow matching (FM) (Lipman et al., 2023; Albergo et al., 2025; Liu et al., 2023) reframes CNF training as a simple regression problem: a vector field is learned to match the endpoint displacement between a prior sample and its paired data point, yielding near-optimal transport trajectories when the prior is Gaussian. However, the independent pairing of FM cannot ensure that the marginal flow follows an optimal transport geodesic, leading to suboptimal paths in practice. In its most general form, conditional flow matching (CFM) (Lipman et al., 2023; Tong et al., 2024) generalizes FM by learning a vector field that transports samples from an arbitrary transport map, conditioned on paired source and target samples. This method allows for simulation-free training of continuous normalizing flows and can learn conditional generative models from any sampleable source distribution, extending beyond the Gaussian source.

Thanks to its flexibility, CFM has been applied in many areas of science, such as molecule generation (Irwin et al., 2024; Geffner et al., 2025), sequence and time-series modeling (Stark et al., 2024; Zhang et al., 2024; Rohbeck et al., 2025), and text-to-speech translation (Guo et al., 2024). A refinement of CFM is minibatch CFM (OT-CFM) (Pooladian et al., 2023; Tong et al., 2024), which uses an entropic or exact plan as the coupling so that each training pair is drawn according to the optimal transport solution between the minibatch source and target samples. This yields substantially straighter, lower-cost trajectories in the learned flow and improves sample quality with fewer integration steps. However, computing these OT plans—even approximately via Sinkhorn—for every minibatch incurs substantial per-iteration overhead, scaling cubically with batch size (or quadratically under entropic regularization). Moreover, requiring well-balanced class representation in each

batch to approximate the global OT plan makes this approach impractical for large, multi-class datasets.

As an alternative that addresses these limitations, we introduce weighted conditional flow matching (W-CFM), which replaces costly batch-level transport computations by simply weighting each independently sampled pair $(x, y)$ with the entropic OT (EOT) Gibbs kernel, $w(x, y) = \exp(-c(x, y)/\varepsilon)$ (Cuturi, 2013). This importance weighting provably recovers the entropic OT (EOT) plan up to a controllable bias in the marginals. As a result, the learned flow follows straight paths without ever explicitly solving an OT problem during training. Moreover, we show that W-CFM matches OT-CFM in the large-batch limit, thereby not incurring any of the batch size-related limitations or any extra costs. In practice, W-CFM delivers straight flows and high-quality samples consistently outperforming CFM and achieving comparable performance to OT-CFM, but with no extra overhead. In short, our contributions are the following:

- We introduce a novel CFM variant that, inspired by EOT, incorporates a Gibbs kernel weight on each sample pair and show that our method serves as a new way to approximate the EOT coupling without any additional cost during training.

- We discuss practical design choices to alleviate the change of the marginals with the new loss and derive sufficient conditions under which this change becomes trivial, so the true marginals are approximately preserved.

- We show that as the batch size grows and assuming that the bias in the marginals remains negligible, W-CFM converges to an entropy-regularized version of OT-CFM, retaining some trajectory straightness without the computational scaling issues associated with the OT plan computations.

- We demonstrate on toy and image-generation benchmarks that W-CFM matches or outperforms existing CFM and OT-CFM methods in sample quality, fidelity, and diversity under a sensible choice of the hyperparameter $\varepsilon$ in the Gibbs kernel.

## 2 BACKGROUND

### 2.1 ENTROPIC OPTIMAL TRANSPORT

We refer the reader to Nutz (2021) for a comprehensive introduction to the topic and only mention the relevant results for our work. For $\mu, \nu \in \mathcal{P}(\mathbb{R}^d)$, we recall the definition of the Kullback–Leibler divergence $D_{\mathrm{KL}}(\mu\|\nu) := \int_{\mathbb{R}^d} \log \frac{d\mu}{d\nu}(x)d\mu(x)$ if $\mu \ll \nu$ and $+\infty$ otherwise. Assume that $\mu, \nu$ have finite first moment. We denote the set of couplings between $\mu$ and $\nu$ by $\Pi(\mu, \nu) := \{\pi \in \mathcal{P}(\mathbb{R}^d \times \mathbb{R}^d) : \pi(\mathbb{R}^d, dy) = \nu(dy), \pi(dx, \mathbb{R}^d) = \mu(dx)\}$. We consider the following entropic optimal transport (EOT) problem with parameter $\varepsilon > 0$:

$$\min_{\pi \in \Pi(\mu, \nu)} \int_{\mathbb{R}^d \times \mathbb{R}^d} c(x, y)d\pi(x, y) + \varepsilon D_{\mathrm{KL}}(\pi\|\mu \otimes \nu), \tag{1}$$

where $c : \mathbb{R}^d \times \mathbb{R}^d \to \mathbb{R}_+$ denotes a cost function, typically $c(x, y) = \|x - y\|$, which is taken such that equation 1 is finite. When $\varepsilon = 0$, we recover the classical Monge–Kantorovich transportation problem of moving a distribution $\mu$ to a distribution $\nu$ by minimizing the transport cost as measured by $c$, whose solution is not necessarily unique; we denote by $\pi^\star \in \Pi(\mu, \nu)$ such a solution. EOT is of significant importance in machine learning and scientific computing (Genevay et al., 2018; Peyré & Cuturi, 2019), as it approximates the original Monge–Kantorovich transport problem and can be solved tractably with Sinkhorn's algorithm (Cuturi, 2013; Altschuler et al., 2017). It is a classical result (see, e.g., Chapter 4 in Peyré & Cuturi (2019)) that solving equation 1 is equivalent to solving the following projection problem $\min_{\pi \in \Pi(\mu, \nu)} D_{\mathrm{KL}}(\pi\|\mathcal{K}_\varepsilon)$, with the Gibbs kernel $\mathcal{K}_\varepsilon(dx, dy) := e^{-c(x, y)/\varepsilon}\mu(dx)\nu(dy)$. A convexity argument can be made to prove that there exists a unique minimizer $\pi_\varepsilon \in \Pi(\mu, \nu)$ of equation 1. More specifically, this projection formulation allows us to write the optimal coupling in terms of $\mathcal{K}_\varepsilon$.

**Theorem 1** (Theorem 4.2 in Nutz (2021)). *If $c(x, y) < \infty$ $\mu \otimes \nu$-almost-surely, for any $\varepsilon > 0$ there exist measurable functions $\phi_\varepsilon, \psi_\varepsilon : \mathbb{R}^d \to \mathbb{R}$, referred to as EOT potentials, such that the EOT plan is given by*

$$\pi_\varepsilon(dx, dy) = \exp\left(\phi_\varepsilon(x) + \psi_\varepsilon(y) - \frac{c(x, y)}{\varepsilon}\right)\mu(dx)\nu(dy). \tag{2}$$

In other words, we have $\pi_\varepsilon(dx, dy) = f_\varepsilon(x) g_\varepsilon(y) \mathcal{K}_\varepsilon(dx, dy)$ for some positive functions $f_\varepsilon(x) = \exp(\phi_\varepsilon(x)), g_\varepsilon(y) = \exp(\psi_\varepsilon(y))$. In particular, the component capturing the dependence in the EOT plan $\pi_\varepsilon$ is exactly given by the Gibbs kernel, which is easy to compute, and one only needs to adjust the marginals independently to obtain $\pi_\varepsilon$. Entropic optimal transport is the static counterpart of the dynamic Schrödinger bridge problem (Léonard, 2013). The corresponding bridge defines a stochastic process whose drift (i.e., deterministic part) minimizes the energy along the path (Gushchin et al., 2023). In other words, the Gibbs kernel appears naturally in EOT, which is a problem whose dynamic counterpart naturally implies minimizing the energy along trajectories, which incentivizes short and straight paths.

## 2.2 Conditional Flow Matching

The flow matching methodology Lipman et al. (2023) is a simulation-free method of training a continuous normalizing flow (Chen et al., 2018), i.e., a smooth vector field $v_\theta(t, x)$, for generating samples from a target distribution $\nu \in \mathcal{P}(\mathbb{R}^d)$ given a source distribution $\mu \in \mathcal{P}(\mathbb{R}^d)$. Assume that there exists a vector field $v_t(x)$ such that the flow given by $dx_t = v_t(x_t)dt$ with initial condition $x_0 \sim \mu$ satisfies $x_1 \sim \nu$. The flow matching loss is given by

$$\mathcal{L}_{\mathrm{FM}}(\theta) \coloneqq \mathbb{E}_{t \sim \mathcal{U}(0,1), X_t \sim p_t} \left[ \|v_\theta(t, X_t) - v_t(X_t)\|^2 \right], \tag{3}$$

where $p_t$ denotes the distribution of $x_t$, i.e. a probability path between $\mu$ and $\nu$. Under technical assumptions, $v_t$ generates $p_t$ if and only if they satisfy the continuity equation

$$\frac{\partial p_t}{\partial t} + \nabla \cdot (p_t v_t) = 0, \quad p_0 = \mu, \quad p_1 = \nu. \tag{4}$$

Upon finding a minimizer of equation 3, one integrates the ODE $d\tilde{x}_t = v_\theta(t, \tilde{x}_t)dt$ from a source sample $\tilde{x}_0 \sim \mu$, so that $\tilde{x}_1$ is approximately distributed according to $\nu$. In its most general form, conditional flow matching (Lipman et al., 2023) replaces the intractable FM loss equation 3 by an equivalent loss involving a conditional vector field $v_t(x \mid z)$ and conditional probability path $p_t(x \mid z)$ satisfying equation 4 between $\mu(dx \mid z)$ and $\nu(dx \mid z)$

$$\mathcal{L}_{\mathrm{CFM}}(\theta; q) \coloneqq \mathbb{E}_{t \sim \mathcal{U}(0,1), Z \sim q} \mathbb{E}_{X_t \sim p_t(\cdot \mid Z)} \left[ \|v_\theta(t, X_t) - v_t(X_t \mid Z)\|^2 \right], \tag{5}$$

where $z$ denotes a latent variable and $q$ the prior distribution, typically choosing $z = x_1$. Tong et al. (2024) have generalized the approach of Lipman et al. (2023) by considering arbitrary latent variables $z$, showing that minimizing the CFM loss equation 5 is equivalent to minimizing equation 3. Hence, the conditional flow matching method calls for two important design choices:

- The latent variable $z$ and prior $q$: in this paper we focus on the popular choice of $z = (x_0, x_1)$, and we therefore require $q$ to be a coupling, that is $q \in \Pi(\mu, \nu)$.
- The conditional vector field $v_t(x \mid z)$ and probability path $p_t(x \mid z)$: in this paper we consider the linear interpolation path $X_t = (1 - t)X_0 + tX_1$, i.e. $p_t(x \mid x_0, x_1) = \delta_{(1-t)x_0 + tx_1}(x)$ together with $v_t(x \mid x_0, x_1) = x_1 - x_0$, this is the implicit choice made for Rectified Flow (Liu et al., 2023).

Choosing $q = \mu \otimes \nu$ in equation 5 leads to the independent conditional flow matching algorithm (I-CFM), which is straightforward to compute but yields irregular trajectories, requiring fine discretizations at inference. On the contrary, choosing $q = \pi^\star$ (i.e., an optimal transport plan between the source and target distributions), leads to the optimal transport conditional flow matching algorithm (OT-CFM), which induces straighter trajectories for the learned model $v_\theta(t, x)$ (Tong et al., 2024). As $\pi^\star$ is *a priori* difficult to sample from, many works have explored approximations by computing short distance couplings at the batch level during training (Tong et al., 2024; Pooladian et al., 2023).

## 3 Weighted Conditional Flow Matching

By writing $L_\theta(t, X, Y) = \|v_\theta(t, X_t) - v_t(X_t \mid X, Y)\|^2$ where $X_t = (1-t)X + tY$, the I-CFM loss can be written as $\mathcal{L}_{\mathrm{I-CFM}}(\theta) = \mathbb{E}_{t \sim \mathcal{U}(0,1), (X,Y) \sim \mu \otimes \nu} [L_\theta(t, X, Y)]$. In order to enforce straightness of the learned velocity field, one would like to bias this loss towards training sample pairs $(x, y)$ that

are close to each other. OT-CFM achieves this by computing the OT plan between discrete sets of points within a batch. We propose to introduce a bias directly inside the expectation by considering modifications of the I-CFM loss function of the form

$$\mathcal{L}_w(\theta) \coloneqq \mathbb{E}_{t \sim \mathcal{U}(0,1), (X,Y) \sim \mu \otimes \nu} \left[ w(X,Y) L_\theta(t,X,Y) \right], \tag{6}$$

where $w : \mathbb{R}^d \times \mathbb{R}^d \to \mathbb{R}_+^*$ denotes a positive weighting function, which should be large whenever $X, Y$ are close and small when $X, Y$ are far apart. This weighting can be understood in terms of a change of measure. In particular, by defining $\pi_w(dx, dy) \propto w(x,y)\mu(dx)\nu(dy)$ we have $\mathcal{L}_w(\theta) = \mathbb{E}_{t \sim \mathcal{U}(0,t), (X,Y) \sim \pi_w} \left[ L_\theta(t,X,Y) \right]$. In other words, weighting the I-CFM loss yields a CFM loss with a new prior distribution $q = \pi_w$. This technique can be thought of as a form of importance sampling, where the baseline distribution $\mu \otimes \nu$ is easy to sample from, and $w(x,y)$ corresponds to the importance sampling weight.

### 3.1 Approximating EOT with Weighted Conditional Flow Matching

Let $\varepsilon > 0$. We propose a new loss function that can be used as a drop-in replacement within conditional flow matching and call it weighted conditional flow matching (W-CFM). Given the form of the EOT plan equation 2, and a cost function $c : \mathbb{R}^d \times \mathbb{R}^d \to \mathbb{R}$, we consider $\mathcal{L}_w$ equation 6 with the weighting function $w_\varepsilon(x,y) \coloneqq \exp(-c(x,y)/\varepsilon)\hat{f}_\varepsilon(x)\hat{g}_\varepsilon(y)$, where $\hat{f}_\varepsilon(x), \hat{g}_\varepsilon(y)$ are independent stochastic estimates of $f_\varepsilon(x)$ and $g_\varepsilon(y)$. We introduce the weighted conditional flow matching loss

$$\mathcal{L}_{\text{W-CFM}}(\theta; \varepsilon) \coloneqq \mathbb{E}_{t \sim \mathcal{U}(0,1)} \mathbb{E}_{(X,Y) \sim \mu \otimes \nu} \left[ w_\varepsilon(X,Y) \| v_\theta(t,X) - (Y-X) \|^2 \right]. \tag{7}$$

**Proposition 1.** *The W-CFM loss defined in equation 7 satisfies $\mathcal{L}_{\text{W-CFM}}(\theta; \varepsilon) = Z_\varepsilon \mathcal{L}_{\text{CFM}}(\theta; q_\varepsilon)$, where $q_\varepsilon$ is the following prior*

$$q_\varepsilon(dx, dy) \coloneqq Z_\varepsilon^{-1} \frac{\mathbb{E}[\hat{f}_\varepsilon(x)]\mathbb{E}[\hat{g}_\varepsilon(y)]}{f_\varepsilon(x)g_\varepsilon(y)} \pi_\varepsilon(dx, dy), \tag{8}$$

*and $Z_\varepsilon$ is the normalizing constant. In particular if for any $x, y$, $\hat{f}_\varepsilon(x)$ and $\hat{g}_\varepsilon(y)$ are unbiased estimates of $f_\varepsilon(x)$ and $g_\varepsilon(y)$ up to constant factors, then, $\mathcal{L}_{\text{W-CFM}}(\theta; \varepsilon) \propto \mathcal{L}_{\text{CFM}}(\theta; \pi_\varepsilon)$ where $\pi_\varepsilon$ is the optimal EOT plan.*

Thus, training a CNF model using the W-CFM loss given by equation 7 is equivalent to training a CNF using the EOT plan as the prior distribution, up to a change (a.k.a. tilt) in the marginals given by the approximation of $f_\varepsilon$ and $g_\varepsilon$. Hence, in the general case $\mathcal{L}_{\text{W-CFM}}$ can be thought of as an approximation of the following loss function

$$\mathcal{L}_{\text{EOT-CFM}}(\theta; \varepsilon) = \mathcal{L}_{\text{CFM}}(\theta; \pi_\varepsilon) = \mathbb{E}_{t \sim \mathcal{U}(0,1)} \mathbb{E}_{(X,Y) \sim \pi_\varepsilon} \left[ \| v_\theta(t, X_t) - (Y-X) \|^2 \right], \tag{9}$$

### 3.2 Marginal Tilting under W-CFM

Using $q_\varepsilon$ for the prior leads to the following tilted marginals, which are obtained by integrating equation 8 with respect to $y$ and $x$ respectively:

$$\tilde{\mu}_\varepsilon(dx) = \tau_{\mu,\varepsilon}(x)\mu(dx), \quad \tilde{\nu}_\varepsilon(dy) = \tau_{\nu,\varepsilon}(y)\nu(dy), \tag{10}$$

Using equation 8, the unnormalized densities of the tilted marginals with respect to the original ones are given by

$$\tau_{\mu,\varepsilon}(x) \coloneqq \frac{d\tilde{\mu}_\epsilon}{d\mu}(x) \propto \int_{\mathbb{R}^d} e^{-\frac{c(x,y)}{\varepsilon}} \mathbb{E}\left[\hat{g}_\varepsilon(y)\right] \nu(dy) \mathbb{E}\left[\hat{f}_\varepsilon(x)\right],$$

$$\tau_{\nu,\varepsilon}(y) \coloneqq \frac{d\tilde{\nu}_\epsilon}{d\nu}(y) \propto \int_{\mathbb{R}^d} e^{-\frac{c(x,y)}{\varepsilon}} \mathbb{E}\left[\hat{f}_\varepsilon(x)\right] \mu(dx) \mathbb{E}[\hat{g}_\varepsilon(y)]. \tag{11}$$

Consequently, training a CNF using the W-CFM loss induces a vector field mapping $\tilde{\mu}_\varepsilon$ to $\tilde{\nu}_\varepsilon$. We formalize this result in the following proposition.

**Proposition 2** (Marginal tilting and continuity equation). *Assume $\mu, \nu \in \mathcal{P}(\mathbb{R}^d)$ have finite second moment. Consider the variational problem*

$$\min_v \; \mathbb{E}_{t \sim \mathcal{U}(0,1)} \, \mathbb{E}_{(X,Y) \sim \mu \otimes \nu} \big[ w_\varepsilon(X,Y) \, \|v(t, X_t) - (Y - X)\|^2 \big], \quad X_t = (1-t)\, X + t\, Y. \quad (12)$$

*Let $\rho_t$ denote the law of $X_t$ under $(X,Y) \sim q_\varepsilon$. Then, equation 12 admits a minimizer $v_\varepsilon \in L^2([0,1] \times \mathbb{R}^d; \rho_t(dx)dt)$, which is unique in that space. Moreover $(\rho, v_\varepsilon)$ solve the continuity equation in the weak sense*

$$\partial_t \rho_t + \nabla \cdot (\rho_t \, v_\varepsilon) = 0, \qquad \rho_0 = \tilde{\mu}_\varepsilon, \; \rho_1 = \tilde{\nu}_\varepsilon.$$

In other words, under mild regularity conditions, the flow generated by $v_\varepsilon$ pushes $\tilde{\mu}_\varepsilon$ forward onto $\tilde{\nu}_\varepsilon$. We now present a way to evaluate the marginal tilting. These densities can be estimated by Monte Carlo sampling. If $\tau_{\mu,\varepsilon}(x)$ is constant $\mu$ almost-everywhere, then one is guaranteed that the source marginal is preserved, i.e., that $\tilde{\mu}_\varepsilon = \mu$. Similarly, if $\tau_{\nu,\varepsilon}(y)$ is constant $\nu$ almost-everywhere, then $\tilde{\nu}_\varepsilon = \nu$. Computing unbiased low-variance approximations of $f_\varepsilon$ and $g_\varepsilon$ is a notorious challenge in EOT, and is typically done using the Sinkhorn algorithm (Cuturi, 2013). We give an example which induces a preservation of the $\nu$ marginal under a naive constant approximation of $\hat{f}_\varepsilon$ and $\hat{g}_\varepsilon$.

**Proposition 3.** *Let $\mathbb{S}_R^{d-1} = \{z \in \mathbb{R}^d : \|z\| = R\}$. Assume $c(x,y) = \|x - y\|$, and that $\mu$ is a rotation-invariant measure, for instance $\mu = \mathcal{N}(0, \sigma^2 I)$. Take $w_\varepsilon(x,y) = \exp(-c(x,y)/\varepsilon)$. Then, $\tau_{\nu,\varepsilon}(y) \equiv C(R, \varepsilon)$ for all $y \in \mathbb{S}_R^{d-1}$, where $C(R, \varepsilon)$ is a nonnegative constant. In particular, if $\nu$ is supported on $\mathbb{S}_R^{d-1}$, then $\tilde{\nu}_\varepsilon = \nu$.*

Proposition 3 implies that using W-CFM with an isotropic distribution as the source, a target distribution which is supported on a $d-1$-dimensional sphere of fixed radius, and trivial approximation of the ratios, will induce a coupling that does not tilt the target marginal. In particular, when using a smooth cost and an appropriate $\varepsilon$, we expect that the target distribution will not be tilted significantly provided that its mass is concentrated on a thin annulus, which typically happens when normalizing high-dimensional data. The proofs of all the above Propositions can be found in Appendix A.

### 3.2.1 On the Choice of $\varepsilon$

The entropy regularization constant $\varepsilon$ controls the trade-off between geometric bias (shorter, straighter flows) and marginal distortion (changing $\mu, \nu$ into $\tilde{\mu}_\varepsilon, \tilde{\nu}_\varepsilon$). As suggested in the previous section, equation 11 can be used to build a proxy to identify sensible values for $\varepsilon$. In particular, if $\tau_{\mu,\varepsilon}, \tau_{\nu,\varepsilon}$ are approximately constant over the supports of $\mu$ and $\nu$ respectively, the tilted marginals $\tilde{\mu}_\varepsilon$ and $\tilde{\nu}_\varepsilon$ remain close to the original ones. In this case, the W-CFM loss in equation 7 closely approximates the EOT-CFM loss in equation 9. To quantify this, we estimate $\tau_{\mu,\varepsilon}$ and $\tau_{\nu,\varepsilon}$ by Monte Carlo sampling over a small number of batches and compute their relative variance, defined as $\frac{\text{Var}(\tau_{\mu,\varepsilon}(X))}{\mathbb{E}[\tau_{\mu,\varepsilon}(X)]^2}$—and analogously for $\tau_{\nu,\varepsilon}$. This metric measures how close the functions are to being constant and is invariant to scaling by a constant factor. This invariance ensures the metric is comparable across datasets, allowing consistent evaluation of how much the importance weights distort the marginals. Low relative variance indicates that the induced marginals are close to $\mu$ and $\nu$, suggesting that the selected $\varepsilon$ yields a good approximation. A formal algorithm describing this heuristic can be found in Appendix D.

As an initial heuristic for choosing $\varepsilon$ in high-dimensional settings (e.g., images or language embeddings) with Euclidean cost $c(x,y) = \|x - y\|$, we rely on the observation that normalized high-dimensional data typically concentrates near a thin spherical shell of radius $\sqrt{d}$, where $d$ is the data dimension (see concentration of measures in Vershynin (2018)). Consequently, the typical inter-sample distance (and hence the typical value of the $L^2$ cost function) is $\mathcal{O}(\sqrt{d})$. Selecting $\varepsilon$ on this same scale ensures that the ratio inside the exponential of the weighting function in equation 7 is $\mathcal{O}(1)$[1]. In this way, the kernel varies slowly with respect to typical data variations since most pairwise distances become comparable to $\varepsilon$. This reasoning is similar to the heuristic commonly used in kernel methods (e.g., SVMs), where the Gaussian kernel width is set proportional to the median pairwise distance between points (Christianini et al., 2000).

---

[1] The same logic applies for different cost functions, e.g., when the cost function is the squared Euclidean norm, then our epsilon should be $O(d)$.

Following this rationale, in all our experiments we set $\varepsilon = \kappa\sqrt{d}$, with the scalar $\kappa$ tuned efficiently (within seconds and a few lines of code) using the relative variance proxy described previously. Specifically, we search over a grid of $\kappa$ values spaced uniformly in log scale and select the smallest value for which the relative variance starts flattening, following an "elbow rule" heuristic akin to the selection of the number of principal components in PCA (Jolliffe, 2002). Across high-dimensional experiments, we consistently find that the optimal values of the constant $\kappa$ satisfy $\kappa << 1$. Additionally, we experimented with various schedulers for $\varepsilon$ (including cosine, exponential, and linear), none of which showed a significant improvement over a fixed constant $\varepsilon$. The exact values of $\varepsilon$ we used are reported along with the results in Section 5.

## 3.3 EQUIVALENCE TO OT-CFM IN THE LARGE BATCH LIMIT

OT–CFM relies on solving a mini-batch optimal transport problem for each batch, which (i) requires batch sizes large enough to represent every mode or class—otherwise the empirical OT plan might be a poor approximation of the true OT plan—and (ii) incurs at least cubic (or quadratic for the entropic approximation) cost in the batch size. By contrast, W-CFM uses a simple and virtually free per-pair Gibbs weight $w_\varepsilon(x, y) = \exp(-c(x,y)/\varepsilon)\hat{f}_\varepsilon(x)\hat{g}_\varepsilon(y)$, avoiding any coupling step. Under regularity assumptions (no marginal tilt, bounded support), one can show that as the batch size goes to infinity, the batch-level EOT-CFM loss converges to a limit which is proportional to the W-CFM loss. We formalize this in Proposition 4 below—proof is given in Appendix A. A more detailed discussion can be found in Appendix B.

**Proposition 4.** *Let $\varepsilon > 0$. Suppose that $\mu, \nu, c$ are such that equation 1 is finite and $\mu, \nu$ have bounded support. Let $(t_n, x_n, y_n)_{n\geq 1}$ be iid samples of $\mathcal{U}(0,1) \otimes \mu \otimes \nu$. Assume that $\tilde{\mu}_\varepsilon = \mu$ and $\tilde{\nu}_\varepsilon = \nu$. Let $\pi_\varepsilon$ be the EOT plan between $\mu$ and $\nu$. Let $\pi_\varepsilon^n$ be the EOT plan between the empirical distributions $\mathbf{x}_n = \frac{1}{n}\sum_{i=1}^{n} \delta_{x_i}$ and $\mathbf{y}_n = \frac{1}{n}\sum_{i=1}^{n} \delta_{y_i}$. Then, $\pi_\varepsilon^n \to \pi_\varepsilon$ almost surely as $n \to \infty$ in the weak sense. In particular, if $\mathcal{B}_n = \{(t_i, x_i, y_i) : 1 \leq i \leq n\}$ and $v_\theta(t, z)$ is uniformly square-integrable in $t \in [0, 1]$, continuous in $z \in \mathbb{R}^d$, we have, for any $\theta$*

$$\mathbb{E}\left[L_{\text{EOT-CFM}}(\mathcal{B}_n, \theta; \varepsilon)\right] \to l(\theta; \varepsilon) \propto \mathcal{L}_{\text{W-CFM}}(\theta; \varepsilon), \text{ as } n \to \infty,$$

*where the expectation is taken over the random batch $\mathcal{B}_n$ and*

$$L_{\text{EOT-CFM}}(\mathcal{B}_n, \theta; \varepsilon) = \frac{1}{n}\sum_{i,j=1}^{n} \pi_\varepsilon^n(x_i, y_j)\|v_\theta(t_i, (1-t_i)x_i + t_i y_j) - (y_j - x_i)\|^2.$$

## 4 RELATED WORK

**Straightening Sample Paths with Optimal Transport.** We review the existing literature on techniques to learn straighter sample paths for flow-based models. Including OT priors within the maximum likelihood training of CNFs has been considered by Onken et al. (2021). Zhang et al. (2025b) propose to learn an acceleration field to capture a random vector field, thereby allowing sample trajectories to cross and leading to straighter trajectories. As highlighted above, our work is closely related to the minibatch optimal transport method proposed by Lipman et al. (2023) and Pooladian et al. (2023) concurrently, which leads to OT-CFM. Within OT-CFM, Klein et al. (2025) have advocated for using the Sinkhorn algorithm (hence computing EOT) to scale to larger batch sizes. Other works have explored an adaptation of OT-CFM to conditional generation through conditional optimal transport (Kerrigan et al., 2024; Cheng & Schwing, 2025). Finally, recent efforts have been made to learn straighter marginal probability paths directly using Wasserstein gradient flows and the JKO scheme (Choi et al., 2024). Compared to OT-CFM, our method has no issue scaling with the batch size, see the discussion in Section 3.3.

**Other Approaches for Faster Inference.** One important line of research has been to distill existing models (Luhman & Luhman, 2021; Salimans & Ho, 2022), either diffusion or flow-based, and learn the corresponding flow maps so that inference can be done in very few steps. Liu et al. (2023) propose to learn straighter trajectories via ReFlow steps, where one trains a student model using couplings generated by a base teacher model trained by flow matching. The ReFlow paradigm is still of interest and continues to be improved Kim et al. (2025). Consistency models Song et al. (2023) were developed to overcome the cost of discretizing the probability flow ODE in diffusion

models, by using the self-consistency property of the underlying flow map. This approach has been translated to flow matching models by Yang et al. (2024). Finally, some great effort has been put into using efficient integrators tailored for these models (Lu et al., 2022; Liu et al., 2022; Zhang & Chen, 2023; Sabour et al., 2024; Williams et al., 2024). Compared to ReFlow, our method does not rely on synthetic data for training and does not require integrating a base neural ODE to generate the training pairs.

**Biasing Flow Matching via Weighting**   Prior works have explored some form of weighted flow matching for different purposes. Energy-weighted conditional flow matching (Zhang et al., 2025a) is motivated by learning a model that directly samples from a tilted target distribution, where the tilting is known beforehand, whereas we try to generate straighter paths for a base model by relying on the a priori unknown EOT plan. Moreover, the energy functional considered by Zhang et al. (2025a) is only applied to the target sample and has little to no effect on the geometry of the path. In our case, we are introducing a weighting scheme that takes as input both endpoints, and leads to a straightening of the paths, while minimizing the tilting. Similarly, flow matching has been used in reinforcement learning to sample from complex policies. The flow matching loss can be computed using a weighting scheme given by the learned advantage function, akin to advantage-weighted regression (Peters & Schaal, 2007; Peng et al., 2019), which biases the model to sample actions with high advantage (Park et al., 2025).

## 5 EXPERIMENTS

We evaluate our flow matching framework across three complementary domains: 2D toy transports, unconditional image generation, and a fidelity and diversity analysis of the generated samples. In these experiments, we demonstrate our framework's performance and competitiveness in comparison to well-established baselines.

### 5.1 EXPERIMENTAL SETUP

To visually probe the benefits of our weighted loss, we design similar low-dimensional transport benchmarks as in Tong et al. (2024). First, we focus on mapping a distribution concentrated on an annulus to a configurable Mixture of Gaussians (MoG). The second setup consists of recovering the moons 2D dataset from a MoG source. We compare W-CFM for different choices of $\varepsilon$ with the cost $c(x, y) = \|x - y\|$ against both OT-CFM and I-CFM, training a two-layer ELU-MLP with 64 hidden units per layer via Adam with a learning rate of $10^{-3}$ for 60,000 iterations with a default batch size of 64. We evaluate sample quality, path straightness, and marginal density estimates using KDE contours. When using W-CFM, the training loss is a sample average of equation 7, rescaled by a Monte-Carlo approximation of $Z_\varepsilon^{-1}$ computed over a single epoch as a preprocessing step. For the models trained with OT-CFM, we use a version where the exact OT plan is computed for each minibatch.

To validate our approach in higher-dimensional settings, we evaluate on CIFAR-10, CelebA64, and ImageNet64-10—a 64×64 version of 10 ImageNet classes (Deng et al., 2009). We use a UNet backbone (Ronneberger et al., 2015) adapted to each dataset: for CIFAR-10, a smaller model with two residual blocks, 64 base channels, and 16×16 attention; for the rest of the datasets, a deeper UNet with three residual blocks, 128 base channels, a [1, 2, 2, 4] channel multiplier, and additional attention at 32×32 for ImageNet64-10, Food20, and Intel. All models are trained with Adam, a learning rate of $2 \times 10^{-4}$, cosine learning rate scheduling with 5,000 warmup steps, and EMA (decay 0.9999), for 400,000 steps using batch sizes of 128 for CIFAR-10, 64 for CelebA and ImageNet64-10, and 48 for Food20 and Intel. Our goal is not to reach state-of-the-art performance, but to compare flow matching variants under matched computational budgets and architectures.

### 5.2 ILLUSTRATIVE COMPARISON ON TOY DATASETS

**Sample Trajectories on Mixture of Gaussians.**   We consider the task of mapping a mixture of many low-variance Gaussians, whose support is concentrated on an annulus, to a target distribution consisting of a mixture of five low-variance Gaussians. We train W-CFM with $\varepsilon = 0.2$ (small $\varepsilon$) and $\varepsilon = 0.4$ (large $\varepsilon$) and use $\hat{f}_\varepsilon = \hat{g}_\varepsilon = 1$. A specific instance where OT-CFM

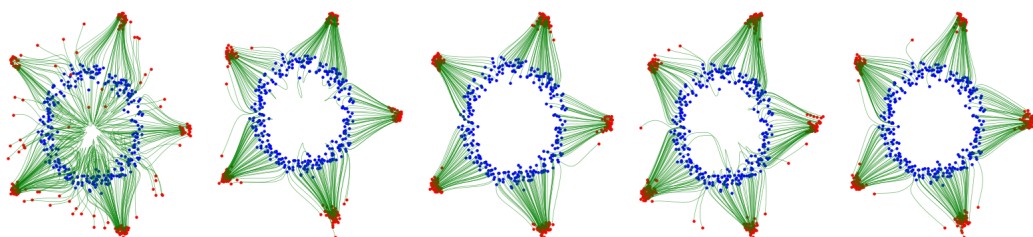

Figure 1: Sample trajectories for circular MoG $\to$ 5 Gaussians. Left to right, the models used are trained with: I-CFM, W-CFM ($\varepsilon = 0.4$), W-CFM ($\varepsilon = 0.2$), OT-CFM (batch size 16), and OT-CFM.

faces challenges is when a typical batch is not representative of the true target distribution (Kornilov et al., 2024; Klein et al., 2025). Hence, on top of training OT-CFM with the default parameters, we train OT-CFM with a smaller batch size to emulate a scenario with a high number of clusters to batch size ratio (we compensate for the smaller batch size by increasing the number of iterations for this version). We plot sample trajectories of the trained models in Figure 1, and report the performance of the models in Table 1. We use the $W_2^2$ distance between generated samples and the true target for overall sample quality, and compute the normalized path energy $\text{NPE}(\theta) = W_2^{-2}(\mu, \nu) \left| \mathbb{E}\left[ \int_0^1 \|v_\theta(t, x_t)\|^2 dt \right] - W_2^2(\mu, \nu) \right|$ for straightness of the trajectories (Tong et al., 2024). Overall, W-CFM leads to better sample quality than OT-CFM with trajectories of similar straightness. We also observe a reduction in straightness for some of the paths when training OT-CFM with a smaller batch size. The W-CFM method outperforms OT-CFM and I-CFM in

Table 1: Comparison of CFM training algorithms' performance on 2D datasets generation on 5 random seeds. $W_2^2$ measures the overall quality of sample generation (lower is better), NPE measures the straightness of trajectories (lower is better), using the true optimal transport cost as a reference. We also report the average time per training iteration in ms, which includes any preprocessing step. **Best** is in bold, second-best is underlined.

| Dataset $\to$ | Circular MoG $\to$ 5 Gaussians | | | 8 Gaussians $\to$ moons | | |
|---|---|---|---|---|---|---|
| Algorithm $\downarrow$ Metric $\to$ | $W_2^2$ ($\downarrow$) | NPE ($\downarrow$) | t/it (ms) | $W_2^2$ ($\downarrow$) | NPE ($\downarrow$) | t/it (ms) |
| I-CFM | $0.091 \pm 0.071$ | $1.703 \pm 0.107$ | 1.274 | $0.680 \pm 0.146$ | $1.033 \pm 0.070$ | **0.894** |
| OT-CFM | $0.029 \pm 0.011$ | **$0.032 \pm 0.019$** | 3.751 | **$0.232 \pm 0.043$** | $0.125 \pm 0.011$ | 1.787 |
| OT-CFM ($B = 16$) | $0.041 \pm 0.014$ | $0.188 \pm 0.041$ | 3.466 | $0.564 \pm 0.125$ | **$0.067 \pm 0.024$** | 1.415 |
| W-CFM (small $\varepsilon$) | **$0.018 \pm 0.008$** | $0.086 \pm 0.021$ | 1.229 | $0.786 \pm 0.324$ | $0.162 \pm 0.068$ | 1.124 |
| W-CFM (large $\varepsilon$) | $0.029 \pm 0.011$ | $0.097 \pm 0.024$ | **1.206** | $0.432 \pm 0.135$ | $0.915 \pm 0.085$ | 1.133 |

terms of sample quality, and it exhibits straightness on par with OT-CFM. In particular, we observe that the trajectories of W-CFM are straighter than those in OT-CFM with a small batch size. Overall, this suggests that our method can be well-suited for unconditional generation involving a target distribution with many clusters.

**Marginal Tilting on Moons Generation.** As highlighted above, our method might induce a tilting of the marginal distributions, requiring a better approximation of $f_\varepsilon$ and $g_\varepsilon$. We investigate this phenomenon when generating moons from a mixture of 8 Gaussians, and training W-CFM with $\varepsilon = 2$ (small $\varepsilon$) and $\varepsilon = 10$ (large $\varepsilon$). We use pre-computed Monte Carlo approximations for $\hat{f}_\varepsilon(x)$ and $\hat{g}_\varepsilon(y)$, as using $\hat{f}_\varepsilon = \hat{g}_\varepsilon = 1$ yields a significant tilting of the marginals (details are given in Appendix C). This is equivalent to performing a single iteration of the Sinkhorn algorithm. As seen in Figure 2, using W-CFM leads to straighter paths compared to I-CFM as measured by NPE, even for a large value of $\varepsilon$, and provides a better sample quality for a careful choice of $\varepsilon$. We present results for $\varepsilon \in \{2, 4, 6, 8, 10\}$ for further validation of this tradeoff in Appendix E.

### 5.3 COMPARISON ON IMAGE DATASETS

To evaluate the generation capabilities of W-CFM beyond low-dimensional toy examples, we conducted unconditional image generation experiments across five different benchmarks: CIFAR-10,

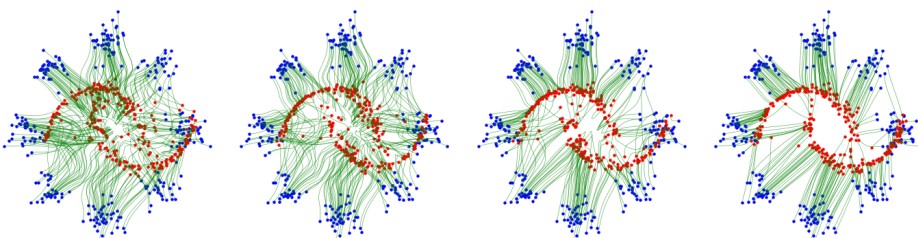

Figure 2: Sample trajectories for moons generation. Source samples are in blue, generated samples are in red. From left to right, we use: I-CFM, W-CFM ($\varepsilon = 10$), W-CFM ($\varepsilon = 2$), and OT-CFM. Here, we use a variant of W-CFM where $\hat{f}_\varepsilon = \hat{f}_{\varepsilon,\mathrm{MC}}, \hat{g}_\varepsilon = \hat{g}_{\varepsilon,\mathrm{MC}}$ (see Appendix C).

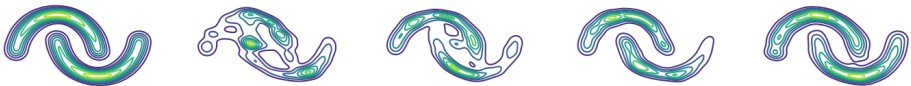

Figure 3: Contour plots of learned density for moons (using 50,000 generated samples). The leftmost plot corresponds to the true target distribution. Then, from left to right, the models used are trained with: I-CFM, W-CFM ($\varepsilon = 10$), W-CFM ($\varepsilon = 2$), and OT-CFM.

CelebA64, ImageNet64-10, Intel Image Classification, and Food20 (a subset of Food101 (Bossard et al., 2014)). Table 2 summarizes the Fréchet Inception Distance (FID, Heusel et al. (2017)) scores obtained by our proposed W-CFM method compared to I-CFM and OT-CFM with an adaptive solver. W-CFM consistently matches or outperforms the baselines, achieving the best FID scores on CIFAR-10, ImageNet64-10, Intel, and Food20, while remaining competitive on CelebA64. The slightly better performance of OT-CFM on CelebA64 is likely explained by the relatively unimodal nature of the dataset (Zhang, 2023), which alleviates the mode representation issues intrinsic to minibatch OT methods (as discussed in Section 3.3). Conversely, the multimodal structure of datasets like CIFAR-10, ImageNet64-10, Intel, and Food20 highlights the advantage of W-CFM. Batch sizes were 128 for CIFAR-10, 64 for ImageNet64-10 and CelebA64, and 48 for the remaining datasets. Applying the proxy described in Section 3.2.1 for selecting $\varepsilon$, we used $\varepsilon = 5$ for CIFAR-10, $\varepsilon = 13$ for CelebA64, $\varepsilon = 14$ for ImageNet64-10, $\varepsilon = 14$ for Intel, and $\varepsilon = 15$ for Food20. We use the naive $\hat{f}_\varepsilon = \hat{g}_\varepsilon = 1$, a choice motivated by the remark at the end of Section 3.2, .

Table 2: FID ↓ (lower is better) across datasets for different flow matching models with Dopri5.

| Model | CIFAR-10 | CelebA64 | ImageNet64-10 | Intel | Food20 |
|-------|----------|----------|---------------|-------|--------|
| I-CFM | 7.44 | 21.99 | 13.86 | 27.54 | 8.15 |
| OT-CFM | 7.60 | **20.93** | 14.39 | 25.63 | 8.23 |
| W-CFM | **7.33** | 21.96 | **13.56** | **25.22** | **7.93** |

We further assessed model efficiency by comparing FID scores at various numbers of neural function evaluations (NFEs) during Euler integration. Table 3 shows that W-CFM consistently achieves lower or comparable FID scores at fewer NFEs compared to both I-CFM and OT-CFM. On CelebA64, OT-CFM outperforms both models, which again reflects the dataset's unimodal structure that mitigates OT-CFM's batch limitations. Appendix E shows example samples for each dataset. Remarkably, although the chosen values of $\varepsilon$ are always an order of magnitude smaller than $\sqrt{d}$, we still obtain good sample quality, suggesting that the marginal tilting is benign in these high-dimensional settings.

## 5.4 FIDELITY AND DIVERSITY OF THE GENERATED SAMPLES

A natural concern with the importance weighting in W-CFM is that it could skew the learned flow toward easier-to-transport pairs, potentially under-representing low-probability modes or degrading sample quality in certain regions of the data manifold. To test this, we evaluate sample fidelity

Table 3: FID ↓ at varying number of neural function evaluations (NFE) using Euler discretization. Each column reports FID at 50, 100, and 120 NFE.

| Dataset | I-CFM | | | OT-CFM | | | W-CFM | | |
|---|---|---|---|---|---|---|---|---|---|
| | **FID @ NFE** | | | | | | | | |
| | 50 | 100 | 120 | 50 | 100 | 120 | 50 | 100 | 120 |
| CIFAR-10 | 10.87 | 9.76 | 8.68 | 11.03 | 9.89 | 8.53 | **10.53** | **9.28** | **8.08** |
| CelebA64 | 29.49 | 25.26 | 24.50 | **27.76** | **23.86** | **22.93** | 29.32 | 25.22 | 24.37 |
| ImageNet64-10 | **14.94** | **13.91** | 13.86 | 15.67 | 14.78 | 14.68 | 15.82 | 14.17 | **13.71** |
| Intel | 26.72 | 26.40 | 26.20 | 25.45 | 25.98 | 24.26 | **25.01** | **24.47** | **24.08** |
| Food20 | 10.10 | 8.98 | 8.85 | 10.16 | 9.17 | 8.95 | **10.01** | **8.97** | **8.57** |

and diversity using the precision-recall-density-coverage (PRDC) suite of metrics (Naeem et al., 2020) on 10,000 generated samples and 5,000 held-out real images per dataset. Precision measures the fraction of generated samples near the real data manifold, recall assesses coverage of the real distribution, and density and coverage estimate support concentration and breadth, respectively. F1 summarizes the trade-off via the harmonic mean of precision and recall.

Table 4: Sample quality and diversity metrics on **ImageNet64-10**.

| Model | Precision (↑) | Recall (↑) | Density (↑) | Coverage (↑) | F1 (↑) |
|---|---|---|---|---|---|
| I-CFM | **0.75** | **0.69** | 0.91 | 0.94 | **0.72** |
| OT-CFM | 0.74 | 0.67 | 0.91 | **0.96** | 0.71 |
| W-CFM | **0.75** | 0.68 | **0.94** | 0.92 | **0.72** |

Table 4 reports the results on ImageNet64-10, showing that W-CFM maintains comparable or better recall, F1, and density than both I-CFM and OT-CFM, without sacrificing coverage or precision. We observe the same qualitative trends across the remaining datasets, with all three methods exhibiting nearly identical PRDC metrics. For completeness, we also include the corresponding tables for CIFAR-10 and CelebA64 in Appendix E. These results confirm that W-CFM does not impair diversity or mode coverage, even when trained with independent pairwise weighting.

## 6 CONCLUSION

In this work, we introduced weighted conditional flow matching (W-CFM), a novel method that leverages insights from entropic optimal transport (EOT) to efficiently improve path straightness and sample quality in continuous normalizing flows. By weighting each training pair using a Gibbs kernel, our approach approximates the EOT plan without incurring the computational cost of OT plan computations and without being limited by the batch size when solving the optimal transport problem. We derived theoretical conditions under which our approximation preserves the original marginals and provide a practical numerical scheme to mitigate the tilting. We establish equivalence with OT-CFM in the large-batch limit, given that the marginals remain unchanged. Empirical evaluations across low-dimensional toy problems, unconditional image generation tasks, and fidelity-diversity analyses confirm that W-CFM achieves performance competitive with OT-CFM while maintaining the computational efficiency of standard CFM.

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

# A PROOFS OF THEORETICAL RESULTS

## A.1 PROOF OF PROPOSITION 1

We assume that, for any $x, y$, $\hat{f}_\varepsilon(x)$ and $\hat{g}_\varepsilon(y)$ are independent integrable random variables. Moreover, we assume there exists measurable functions $F_\varepsilon$ and $G_\varepsilon$ such that $F_\varepsilon(x) = \mathbb{E}[\hat{f}_\varepsilon(x)], G_\varepsilon(y) = \mathbb{E}[\hat{g}_\varepsilon(y)]$ and we assume that for $X \sim \mu, Y \sim \nu$, $F_\varepsilon(X), G_\varepsilon(Y)$ are integrable random variables. Then, by denoting $\mathbb{E}$ the expectation taken with respect to $t \sim \mathcal{U}(0,1), (X, Y) \sim \mu \otimes \nu$ and the randomness of $\hat{f}_\varepsilon, \hat{g}_\varepsilon$

$$\mathcal{L}_{\text{W-CFM}}(\theta; \varepsilon) = \mathbb{E}\left[w_\varepsilon(X, Y) \|v_\theta(t, X) - (Y - X)\|^2\right]$$

$$= \mathbb{E}\left[\exp(-c(X, Y)/\varepsilon)\hat{f}_\varepsilon(X)\hat{g}_\varepsilon(Y) \|v_\theta(t, X) - (Y - X)\|^2\right]$$

$$= \mathbb{E}\left[\mathbb{E}\left[\hat{f}_\varepsilon(X)\hat{g}_\varepsilon(Y) \mid X, Y\right] \exp(-c(X, Y)/\varepsilon) \|v_\theta(t, X) - (Y - X)\|^2\right].$$

Now, we have

$$\mathbb{E}\left[\hat{f}_\varepsilon(X)\hat{g}_\varepsilon(Y) \mid X, Y\right] = \varphi(X, Y)$$

$$\text{where } \varphi(x, y) = \mathbb{E}\left[\hat{f}_\varepsilon(x)\hat{g}_\varepsilon(y)\right] = \mathbb{E}\left[\hat{f}_\varepsilon(x)\right] \mathbb{E}\left[\hat{g}_\varepsilon(y)\right] = F_\varepsilon(x)G_\varepsilon(y).$$

Hence

$$\mathcal{L}_{\text{W-CFM}}(\theta; \varepsilon) = \mathbb{E}\left[\exp(-c(X, Y)/\varepsilon)F_\varepsilon(X)G_\varepsilon(Y) \|v_\theta(t, X_t) - (Y - X)\|^2\right].$$

We define the following probability measure

$$q_\varepsilon(dx, dy) := Z_\varepsilon^{-1} \frac{F_\varepsilon(x)G_\varepsilon(y)}{f_\varepsilon(x)g_\varepsilon(y)} \pi_\varepsilon(dx, dy) = Z_\varepsilon^{-1} \exp(-c(x, y)/\varepsilon)F_\varepsilon(x)G_\varepsilon(y)\mu(dx)\nu(dy).$$

By a change of measure, we get

$$\mathcal{L}_{\text{W-CFM}}(\theta; \varepsilon) = Z_\varepsilon \mathbb{E}_{t \sim \mathcal{U}(0,1),(X,Y) \sim q_\varepsilon}\left[\|v_\theta(t, X) - (Y - X)\|^2\right] = Z_\varepsilon \mathcal{L}_{\text{CFM}}(\theta; q_\varepsilon),$$

which ends the proof. $\qquad\square$

## A.2 PROOF OF PROPOSITION 2

Recall the prior defined in equation 8. Recall that $\rho_t$ denotes the distribution of $X_t = (1-t)X + tY$ under $(X, Y) \sim q_\varepsilon$. For any $v \in L^2([0, 1] \times \mathbb{R}^d; \rho_t(dx)dt)$, we have

$$\mathbb{E}_{t \sim \mathcal{U}(0,1)}\mathbb{E}_{(X,Y) \sim \mu \otimes \nu}\left[w_\varepsilon(X, Y)\|v(t, X_t) - (Y - X)\|^2\right]$$

$$= \mathbb{E}_{t \sim \mathcal{U}(0,1)}\mathbb{E}_{(X,Y) \sim q_\varepsilon}\left[\frac{d(\mu \otimes \nu)}{dq_\varepsilon}(X, Y) \exp(-c(X, Y)/\varepsilon)\|v(t, X_t) - (Y - X)\|^2\right]$$

$$= Z_\varepsilon \mathbb{E}_{t \sim \mathcal{U}(0,1)}\mathbb{E}_{(X,Y) \sim q_\varepsilon}\left[\|v(t, X_t) - (Y - X)\|^2\right],$$

where $Z_\varepsilon$ denotes the normalizing constant $Z_\varepsilon := \mathbb{E}_{(X,Y) \sim \mu \otimes \nu}[w_\varepsilon(X, Y)] > 0$. Hence the variational problem given by equation 12 is equivalent to

$$\min_v \mathbb{E}_{t \sim \mathcal{U}(0,1)}\mathbb{E}_{(X,Y) \sim q_\varepsilon}\left[\|v(t, X_t) - (Y - X)\|^2\right]. \tag{13}$$

By the $L^2$-projection property of conditional expectations, the variational problem of equation 13 is solved by the function $v_\varepsilon : [0, 1] \times \mathbb{R}^d \to \mathbb{R}^d$ defined by

$$v_\varepsilon(t, z) = \mathbb{E}_{(X,Y) \sim q_\varepsilon}[Y - X \mid X_t = z]. \tag{14}$$

Note that this definition is unique in $L^2([0, 1] \times \mathbb{R}^d; \rho_t(dx)dt)$. We now check that $v_\varepsilon$ generates a valid probability path between $\tilde{\mu}_\varepsilon$ and $\tilde{\nu}_\varepsilon$, i.e., that $(\rho, v_\varepsilon)$ satisfy the continuity equation equation 4

in the weak sense. Clearly, $v_\varepsilon(t, \cdot) \in L^1(\mathbb{R}^d, \rho_t)$ and $\int_0^1 \int_{\mathbb{R}^d} |v_\varepsilon(t, x)| p(t, x) dx dt < \infty$. By Proposition 4.2 in Santambrogio (2015), it is enough to check that the continuity equation is satisfied in the sense of distributions. Let $\phi \in C_c^1((0, 1) \times \mathbb{R}^d)$, then

$$\int_0^1 \int_{\mathbb{R}^d} \partial_t \phi(t, x) \rho_t(dx) dt + \int_0^1 \int_{\mathbb{R}^d} \nabla \phi(t, x) \cdot v_\varepsilon(t, x) \rho_t(dx) dt$$

$$= \int_0^t \mathbb{E}[\partial_t \phi(t, X_t) + \nabla \phi(t, X_t) \cdot (Y - X)] dt = \mathbb{E}[\phi(1, Y) - \phi(0, X)] = 0.$$

$\square$

### A.3 PROOF OF PROPOSITION 3

Let $y_2, y_2 \in \mathbb{S}_R^{d-1}$. We can get hold of $\varphi \in O(d)$ (i.e. a distance-preserving transformation in $\mathbb{R}^d$) such that $\varphi(y_1) = y_2$. Since $\|\varphi(y_1) - \varphi(x)\| = \|y_1 - x\|$ for all $x \in \mathbb{R}^d$, we have

$$\int_{\mathbb{R}^d} \exp\left(-\frac{\|y_2 - x\|}{\varepsilon}\right)) \mu(dx) = \int_{\mathbb{R}^d} \exp\left(-\frac{\|\varphi(y_1) - x\|}{\varepsilon}\right)) \mu(dx)$$

$$= \int_{\varphi^{-1}(\mathbb{R}^d)} \exp\left(-\frac{\|\varphi(y_1) - \varphi(x)\|}{\varepsilon}\right) (\varphi_\#^{-1} \mu)(dx)$$

$$= \int_{\varphi^{-1}(\mathbb{R}^d)} \exp\left(-\frac{\|y_1 - x\|}{\varepsilon}\right) (\varphi_\#^{-1} \mu)(dx)$$

$$= \int_{\mathbb{R}^d} \exp\left(-\frac{\|y_1 - x\|}{\varepsilon}\right) \mu(dx),$$

which directly implies that $\tau_{\mu,\varepsilon}(y_2) = \tau_{\mu,\varepsilon}(y_1)$

$\square$

### A.4 PROOF OF PROPOSITION 4

Let $\varepsilon > 0$. Recall that $(t_n, x_n, y_n)_{n \geq 1}$ are iid samples of $\mathcal{U}(0, 1) \otimes \mu \otimes \nu$, and that we assume $\tilde{\mu}_\varepsilon = \mu$ and $\tilde{\nu}_\varepsilon = \nu$. Let $\pi_\varepsilon$ be the optimal EOT plan between $\mu$ and $\nu$. Let $\pi_\varepsilon^n$ be the optimal EOT plan betwen $\mathbf{x}_n = \frac{1}{n} \sum_{i=1}^n \delta_{x_i}$ and $\mathbf{y}_n = \frac{1}{n} \sum_{i=1}^n \delta_{y_i}$. In this proof, the convergence of probability measures is understood in the weak sense.

First, the almost-sure convergences $\mathbf{x}_n \to \mu$ and $\mathbf{y}_n \to \nu$ come from a classical result in probability theory on the convergence of empirical distributions to the true distribution, see Varadarajan (1958).

Since the minimization problem of equation 1 is non-trivial, an application of Theorem 1.4 in Ghosal et al. (2022) shows that the empirical EOT plan satisfies $\pi_\varepsilon^n \to \pi_\varepsilon$ almost surely. Now, for any $n \geq 1$, we have

$$\mathbb{E}\left[L_{\text{EOT-CFM}}(\mathcal{B}_n, \theta; \varepsilon)\right] = \mathbb{E}\left[\int_{\mathbb{R}^d \times \mathbb{R}^d} \int_0^1 \|v_\theta(t, (1-t)x + ty) - (y - x)\|^2 dt \pi_\varepsilon^n(dx, dy)\right]$$

$$= \mathbb{E}\left[\int_{s(\mu) \times s(\nu)} \int_0^1 \|v_\theta(t, (1-t)x + ty) - (y - x)\|^2 dt \pi_\varepsilon^n(dx, dy)\right],$$

where $s(\mu), s(\nu)$ denote the support of $\mu$ and $\nu$ respectively, which are assumed to be bounded. Since $(x, y) \mapsto \int_0^1 \|v_\theta(t, (1-t)x + ty) - (y - x)\|^2 dt$ is continuous and bounded on $s(\mu) \times s(\nu)$ by our assumption on $v_\theta$, we have

$$\int_{s(\mu) \times s(\nu)} \left(\int_0^1 \|v_\theta(t, (1-t)x + ty) - (y - x)\|^2 dt\right) \pi_\varepsilon^n(dx, dy)$$

$$\to \int_{s(\mu) \times s(\nu)} \left(\int_0^1 \|v_\theta(t, (1-t)x + ty) - (y - x)\|^2 dt\right) \pi_\varepsilon(dx, dy)$$

almost surely as $n \to \infty$. Now, by uniform integrability, this convergence also holds in expectation, i.e.

$$\mathbb{E}\left[L_{\text{EOT-CFM}}(\mathcal{B}_n, \theta; \varepsilon)\right] \to \int_{\mathrm{s}(\mu) \times \mathrm{s}(\nu)} \left(\int_0^1 \|v_\theta(t, (1-t)x + ty) - (y-x)\|^2 \, dt\right) \pi_\varepsilon(dx, dy).$$

Finally, we want to prove that this integral is proportional to $\mathcal{L}_{\text{W-CFM}}(\theta; \varepsilon)$. Since we assume no tilting of the marginals, i.e. $q_\varepsilon = \pi_\varepsilon$, we have

$$\mathcal{L}_{\text{W-CFM}}(\theta; \varepsilon) = Z_\varepsilon \mathbb{E}_{t \sim \mathcal{U}(0,1), (X,Y) \sim \pi_\varepsilon}\left[\|v_\theta(t, X_t) - (Y-X)\|^2\right]$$

$$= Z_\varepsilon \int_{\mathrm{s}(\mu) \times \mathrm{s}(\nu)} \left(\int_0^1 \|v_\theta(t, (1-t)x + ty) - (y-x)\|^2 \, dt\right) \pi_\varepsilon(dx, dy).$$

by using the same change of measure argument as in the proof of Proposition 2. $\qquad\square$

## B   DETAILS ON THE EQUIVALENCE TO OT-CFM IN THE LARGE BATCH LIMIT

We recall the mini-batch optimal transport technique that is central in the OT-CFM algorithm of Tong et al. (2024). Given a batch of i.i.d. samples $\mathcal{B} = \{(t_i, x_i, y_i) : i = 1, \ldots, B\}$, where $t_i$ are i.i.d. according to $\mathcal{U}(0,1)$, $x_i$ are i.i.d. according to $\mu$, $y_i$ are i.i.d. according to $\nu$, and $t_i, x_i, y_i$ are drawn independently, one can compute the optimal transport plan between the two corresponding discrete distribution, i.e. one computes

$$\pi_\mathcal{B} \in \arg\min_{\pi \in \Pi_\mathcal{B}} \sum_{i=1}^B \sum_{j=1}^B c(x_i, y_j) \pi(x_i, y_j), \tag{15}$$

where $\Pi_\mathcal{B}$ is the set of couplings between the empirical measures

$$\mathbf{x}_\mathcal{B} = \frac{1}{B} \sum_{i=1}^B \delta_{x_i}, \quad \mathbf{y}_\mathcal{B} = \frac{1}{B} \sum_{i=1}^B \delta_{y_i}.$$

In particular, any $\pi \in \Pi_\mathcal{B}$ must satisfy $\sum_j \pi(x_i, y_j) = \sum_i \pi(x_i, y_j) = \frac{1}{B}$. Then, given an optimal $\pi_\mathcal{B}$, one computes the following

$$L_{\text{OT-CFM}}(\mathcal{B}, \theta) = \frac{1}{B} \sum_{i=1}^B (v_\theta(t_i, (1-t_i)x_i + t_i y_{\sigma(i)}) - (y_{\sigma(i)} - x_i))^2, \tag{16}$$

where $\sigma$ is a permutation corresponding to a Monge map for the problem equation 15, i.e., for some $T : \{x_i : i = 1, \ldots, B\} \to \{y_i : i = 1, \ldots, B\}$ such that $T(x_i) = y_{\sigma(i)}$ and $\pi_\mathcal{B} := (\text{Id}, T)_\# \mathbf{x}_\mathcal{B}$ is a solution to equation 15 (Peyré & Cuturi, 2019). This sample loss is used as an approximation of the following OT-CFM loss

$$\mathcal{L}_{\text{OT-CFM}}(\theta) := \mathbb{E}_{t \sim \mathcal{U}(0,1)} \mathbb{E}_{(X,Y) \sim \pi^\star}\left[\|v_\theta(t, X_t) - (Y-X)\|^2\right], \tag{17}$$

where $\pi^\star$ solves the unregularized optimal transport problem, that is equation 1 with $\varepsilon = 0$.

The sample OT-CFM loss in equation 16 is a low bias approximation of equation 17 only when the batch size is large enough. The actual samples for which we compute equation 16 are not exactly distributed according to a genuine OT plan between $\mu$ and $\nu$, since the OT plan $\pi^\star$ and the product measures $\mu \otimes \nu$ might be mutually singular. Additionally, computing the exact batch OT plan becomes prohibitively expensive as the batch size grows. A solution is to compute an approximate OT plan, by using the Sinkhorn algorithm (Cuturi, 2013), which is an efficient way of computing the entropic OT plan between two discrete sets of measures. In that case, as the batch size increases, the sample OT-CFM loss equation 16 approximates $\mathcal{L}_{\text{EOT-CFM}}$ given by equation 9. Nevertheless, approximating the OT at the batch level is particularly challenging in datasets with multiple modes, as it becomes unrealistic to faithfully approximate the global OT if not all modes are adequately represented within each (or the average) batch. Consequently, the batch size must scale with the number of models or classes present in the dataset.

Our method does not have these scaling issues with the batch size, since it only involves computing a simple weighting factor $w_\varepsilon(x_i, y_i) = \exp(-c(x_i, y_i)/\varepsilon)$ for every training sample pair $(x_i, y_i)$ in a batch. In other words, if one assumes that the weight does not tilt the marginals, the weighted CFM method corresponds to a large batch limit of OT-CFM (where batch EOT is used).

## C  MONTE CARLO ESTIMATES FOR $\hat{f}_\varepsilon$ AND $\hat{g}_\varepsilon$

### C.1  GENERAL METHOD

Whenever the tilting is prominent, as in the 8 Gaussians $\rightarrow$ moons task, using $\hat{f}_\varepsilon = g_\varepsilon = 1$ is no longer sufficient. Instead, we propose the following scheme to compute Monte Carlo estimates $\hat{f}_{\varepsilon,\mathrm{MC}}, \hat{g}_{\varepsilon,\mathrm{MC}}$ as a pre-processing step:

- First, discretize the source distribution, i.e. draw $N_\mu$ samples $x_1, \ldots, x_{N_\mu}$ from the source distribution. Similarly, draw $N_\nu$ samples $y_1, \ldots, y_{N_\nu}$ from the target distribution (when either distribution is already discretized, as is the case for the image data in most unconditional image generation tasks, just use all the samples available).
- Then, compute

$$\hat{f}_{\varepsilon,\mathrm{MC}}(x_i) = \left( \frac{1}{N_\nu} \sum_{j=1}^{N_\nu} \exp(-c(x_i, y_j)/\varepsilon) \right)^{-1}, \quad i = 1, \ldots, N_\mu,$$

$$\hat{g}_{\varepsilon,\mathrm{MC}}(y_i) = \left( \frac{1}{N_\mu} \sum_{i=1}^{N_\mu} \exp(-c(x_i, y_j)/\varepsilon) \right)^{-1}, \quad j = 1, \ldots, N_\nu,$$

  that is, for every point in the discretized source (resp. every point in the discretized target) compute a Monte Carlo approximation proportional to the true weight $f_\varepsilon$ (resp. $g_\varepsilon$) using all the points in the discretized target (resp. using all the points in the discretized source).

Then, during training, sample pairs of point independently from the discretized distributions, and use the weight $w_\varepsilon(x_i, y_j) = \exp(-c(x_i, y_j)/\varepsilon)\hat{f}_{\varepsilon,\mathrm{MC}}(x_i)\hat{g}_{\varepsilon,\mathrm{MC}}(y_j)$.

### C.2  APPLICATION TO MOONS GENERATION

In order to assess the effectiveness of the scheme described above, we plot trajectories in Figure 4 and report performance in terms of squared 2-Wasserstein distance and NPE in Table 5. We train using W-CFM with different values of $\varepsilon$, and for each value of $\varepsilon$, we train a model with naive $\hat{f}_\varepsilon = \hat{g}_\varepsilon = 1$ and compare against MC estimates $\hat{f}_\varepsilon = \hat{f}_{\varepsilon,\mathrm{MC}}, \hat{g}_\varepsilon = \hat{g}_{\varepsilon,\mathrm{MC}}$. As measured by $W_2^2$, the MC approach yields a significant improvement over the naive approach accross all considered values of $\varepsilon$. Here, the validity of NPE as a measure of path straightness is arguable, especially for the small $\varepsilon$ naive variant, as the generated distribution is significantly tilted.

Table 5: Comparison of variants of W-CFM on **8 Gaussians** $\rightarrow$ **moons** on 5 random seeds. $W_2^2$ measures the overall quality of sample generation (lower is better), NPE measures the straightness of trajectories (lower is better), using the true optimal transport cost as a reference. For each $\varepsilon$, we compare the naive approach ($\hat{f}_\varepsilon = \hat{g}_\varepsilon = 1$) against the MC approach ($\hat{f}_\varepsilon = \hat{f}_{\varepsilon,\mathrm{MC}}, \hat{g}_\varepsilon = \hat{g}_{\varepsilon,\mathrm{MC}}$)

| Variant $\rightarrow$ | Naive $\hat{f}_\varepsilon = \hat{g}_\varepsilon = 1$ | | MC $\hat{f}_\varepsilon = \hat{f}_{\varepsilon,\mathrm{MC}}, \hat{g}_\varepsilon = \hat{g}_{\varepsilon,\mathrm{MC}}$ | |
| :--- | :---: | :---: | :---: | :---: |
| Algorithm $\downarrow$ Metric $\rightarrow$ | $W_2^2$ ($\downarrow$) | NPE ($\downarrow$) | $W_2^2$ ($\downarrow$) | NPE ($\downarrow$) |
| W-CFM ($\varepsilon = 2$) | $1.823 \pm 0.166$ | $0.289 \pm 0.008$ | $0.786 \pm 0.324$ | $0.162 \pm 0.068$ |
| W-CFM ($\varepsilon = 4$) | $1.476 \pm 0.167$ | $0.033 \pm 0.023$ | $0.564 \pm 0.337$ | $0.439 \pm 0.105$ |
| W-CFM ($\varepsilon = 6$) | $0.960 \pm 0.186$ | $0.220 \pm 0.050$ | $0.484 \pm 0.128$ | $0.627 \pm 0.118$ |
| W-CFM ($\varepsilon = 8$) | $0.888 \pm 0.217$ | $0.365 \pm 0.076$ | $0.562 \pm 0.181$ | $0.833 \pm 0.031$ |
| W-CFM ($\varepsilon = 10$) | $0.843 \pm 0.321$ | $0.463 \pm 0.061$ | $0.432 \pm 0.135$ | $0.915 \pm 0.085$ |

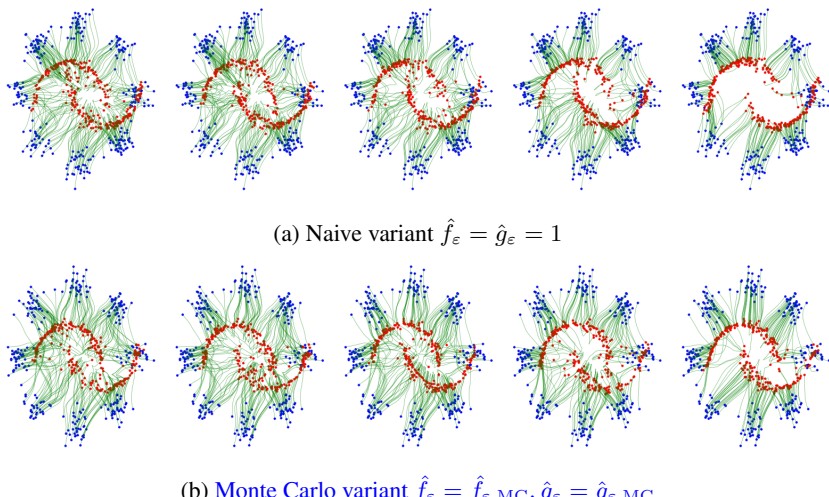

(a) Naive variant $\hat{f}_\varepsilon = \hat{g}_\varepsilon = 1$

(b) Monte Carlo variant $\hat{f}_\varepsilon = \hat{f}_{\varepsilon,\mathrm{MC}}, \hat{g}_\varepsilon = \hat{g}_{\varepsilon,\mathrm{MC}}$

Figure 4: Sample trajectories on **8 Gaussians** $\rightarrow$ **moons** with different variants of W-CFM. From left to right, the models used are trained with the following values of $\varepsilon$: 10,8,6,4,2.

## D   ALGORITHM FOR DETERMINING $\varepsilon$

---

**Algorithm 1** Choosing $\varepsilon$ by controlling the variability of the kernel-induced marginal tilts

---

**Require:** Dataset $\{y_j\}_{j=1}^N \sim \nu$, prior sampler for $\mu$, number of source points $M$, candidate scales $\{\varepsilon_\ell\}_{\ell=1}^L$

1: **for** $\ell = 1, \ldots, L$ **do**                                      $\triangleright$ Evaluate one candidate $\varepsilon_\ell$
2:     Sample $\{x_i\}_{i=1}^M \sim \mu$
3:     Define Gibbs kernel

$$K_{ij}^{(\ell)} = \exp\Big( - \tfrac{c(x_i,y_j)}{\varepsilon_\ell} \Big), \qquad 1 \leq i \leq M,\ 1 \leq j \leq N$$

4:     Monte Carlo estimates of the marginal tilting factors in equation 11 (with $\hat{f}_\varepsilon = \hat{g}_\varepsilon \equiv 1$):

$$\widehat{\tau}_{\mu,\varepsilon_\ell}(x_i) = \frac{1}{N} \sum_{j=1}^N K_{ij}^{(\ell)}, \qquad \widehat{\tau}_{\nu,\varepsilon_\ell}(y_j) = \frac{1}{M} \sum_{i=1}^M K_{ij}^{(\ell)}.$$

5:     Compute relative variances of the empirical tilts

$$\mathrm{RV}_\mu(\varepsilon_\ell) = \frac{\mathrm{Var}_i[\widehat{\tau}_{\mu,\varepsilon_\ell}(x_i)]}{(\mathbb{E}_i[\widehat{\tau}_{\mu,\varepsilon_\ell}(x_i)])^2}, \qquad \mathrm{RV}_\nu(\varepsilon_\ell) = \frac{\mathrm{Var}_j[\widehat{\tau}_{\nu,\varepsilon_\ell}(y_j)]}{(\mathbb{E}_j[\widehat{\tau}_{\nu,\varepsilon_\ell}(y_j)])^2}.$$

6: **end for**
7: Choose $\varepsilon^\star$ according to a selection rule, e.g.

$$\varepsilon^\star = \min\Big\{\varepsilon_\ell\ :\ \mathrm{RV}_\mu(\varepsilon_\ell) \leq \delta\ \text{ and } \ \mathrm{RV}_\nu(\varepsilon_\ell) \leq \delta\Big\},$$

for some tolerance $\delta$, or via an "elbow" in the curves $\mathrm{RV}_\mu(\varepsilon_\ell), \mathrm{RV}_\nu(\varepsilon_\ell)$.
8: **return** $\varepsilon^\star$

---

Using Algorithm 1, we determined the values of $\varepsilon$ used in the high-dimensional experiments in Section 5. In particular, we set the tolerance threshold $\delta$ for the relative variance to be on the order of $10^{-2}$, confirming that, for these datasets, the potentials are indeed approximately constant.

# E ADDITIONAL RESULTS

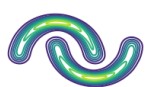 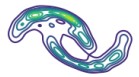 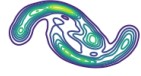 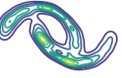 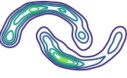 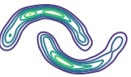

Figure 5: Contour plots of learned target density for **8 Gaussians** → **moons**. The leftmost plot corresponds to the true target distribution. Then, from left to right, the models used are trained with the following values of $\varepsilon$: 10,8,6,4,2.

| $\varepsilon$ | FID $\downarrow$ |
|---|---|
| 1.0 | 14.01 |
| 2.0 | 8.89 |
| 3.0 | 7.65 |
| 5.0 | **7.33** |
| 7.5 | 7.42 |

Table 6: FID scores on CIFAR-10 for different $\varepsilon$.

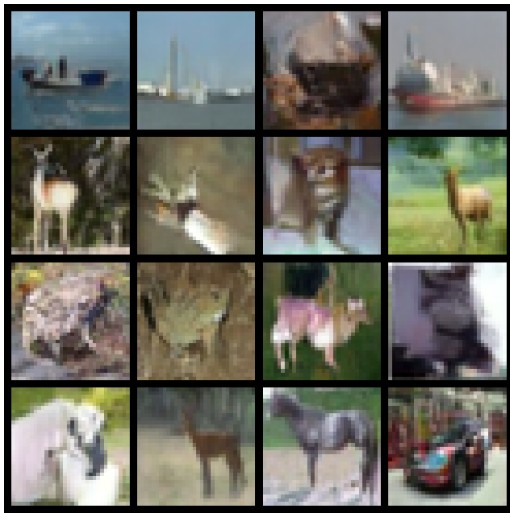

Figure 6: Generated samples from W-CFM trained on CIFAR-10.

Table 7: Sample quality and diversity metrics on **CIFAR-10**.

| Model | Precision ($\uparrow$) | Recall ($\uparrow$) | Density ($\uparrow$) | Coverage ($\uparrow$) | F1 ($\uparrow$) |
|---|---|---|---|---|---|
| I-CFM | **0.83** | 0.75 | 0.98 | 0.91 | **0.78** |
| OT-CFM | 0.80 | 0.75 | **1.00** | **0.92** | 0.77 |
| W-CFM | 0.81 | **0.76** | 0.94 | 0.91 | **0.78** |

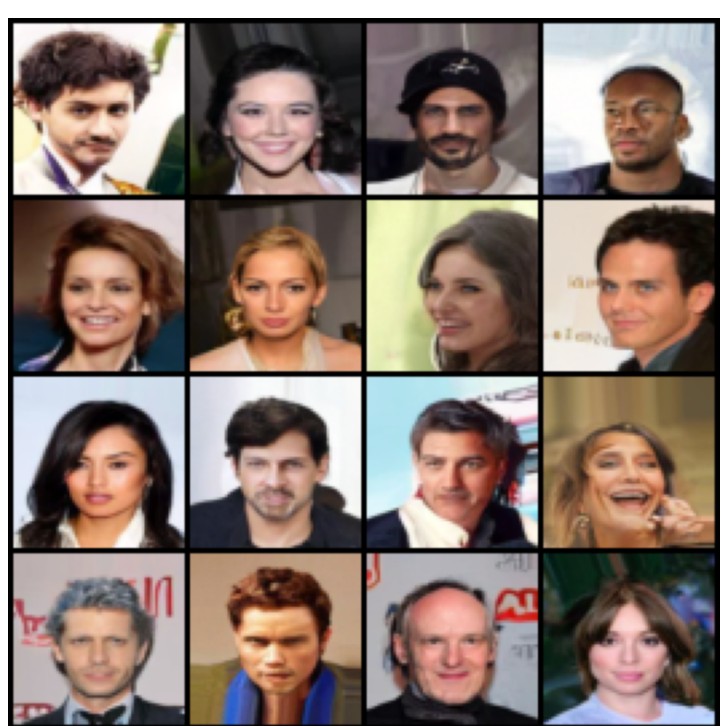

Figure 7: Generated samples from W-CFM trained on CelebA64.

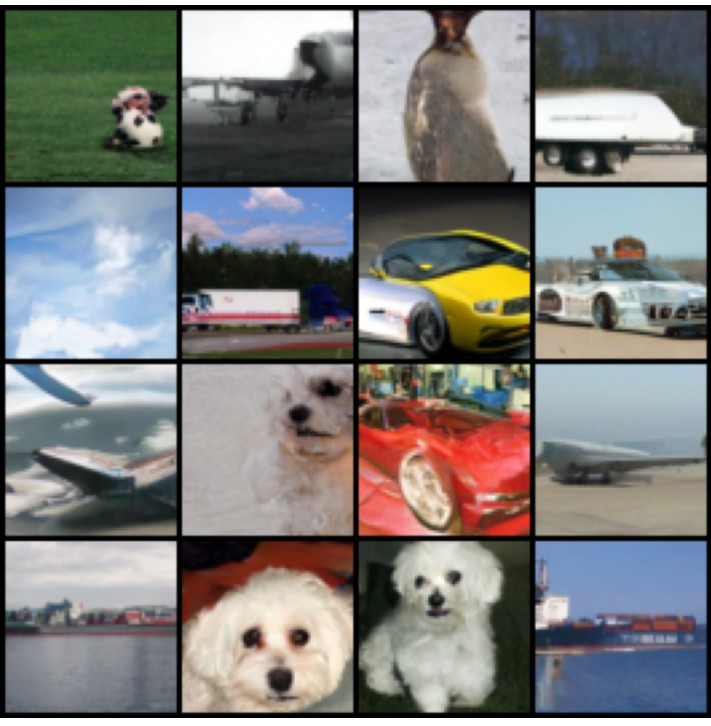

Figure 8: Generated samples from W-CFM trained on ImageNet-10.

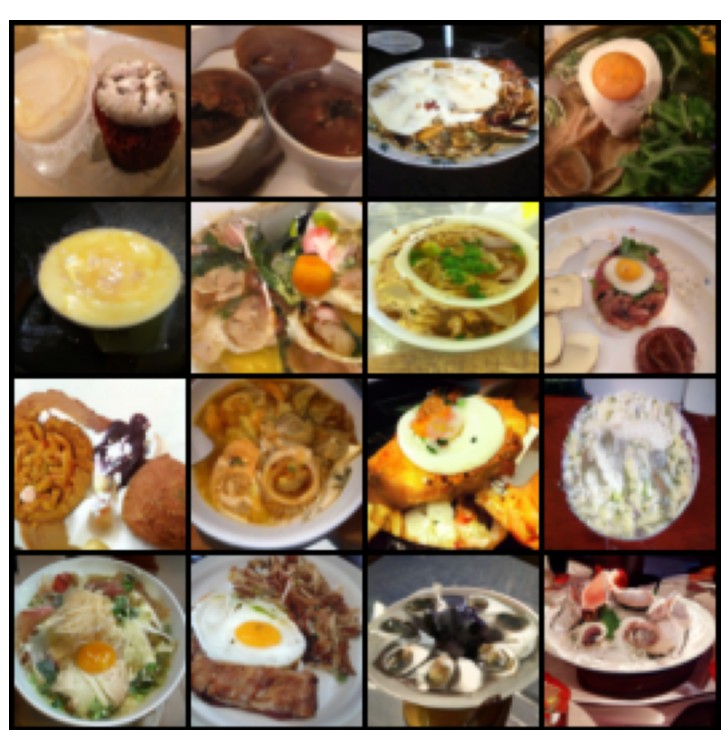

Figure 9: Generated samples from W-CFM trained on Food-101.

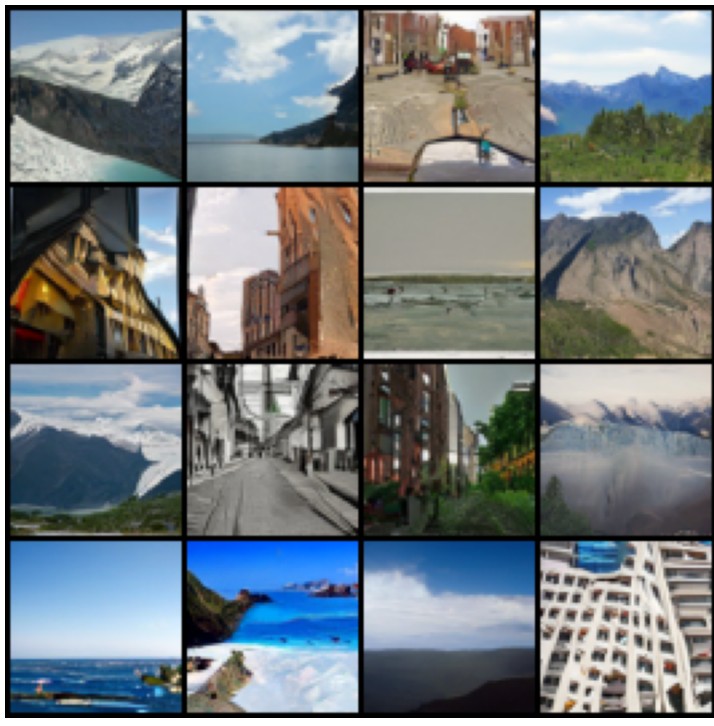

Figure 10: Generated samples from W-CFM trained on Intel Image Classification.

Table 8: Sample quality and diversity metrics on **CelebA64**.

| Model | Precision (↑) | Recall (↑) | Density (↑) | Coverage (↑) | F1 (↑) |
|-------|---------------|------------|-------------|--------------|--------|
| I-CFM | **0.86** | **0.66** | **1.26** | **0.98** | **0.74** |
| OT-CFM | 0.84 | 0.65 | 1.23 | 0.96 | 0.73 |
| W-CFM | 0.83 | **0.66** | 1.19 | **0.98** | **0.74** |

