# OpenReview forum: "Weighted Conditional Flow Matching"
_ICLR.cc/2026/Conference — Submitted to ICLR 2026_

### Official Review · Reviewer_MqhE · 2025-10-26

**Soundness:** 3
**Presentation:** 3
**Contribution:** 2
**Rating:** 6
**Confidence:** 4

**Summary:**

To overcome the computational burden of OT-CFM, this paper proposes to weight the FM loss by per-pair Gibbs weight $w_{\epsilon}(x,y)$ and it leads to the W-CFM. The W-CFM loss approximates the entropic OT up to marginal tilting. They also show large-batch equivalence to OT-CFM. On the choice of tuning parameter $\epsilon$, the paper discuss a huristic way. By simulation and experiements on image datasets, they show the W-CFM has competitive or better FID and path straightness.

**Strengths:**

1. The paper is well written and easy to follow.
2. The training of W-CFM is as cheap as I-CFM, while recovering OT-like straightness.
3. The formal link to EOT and marginal tilt validate the weighting on loss.

**Weaknesses:**

The choice of $\epsilon$ can still be relatively cumbersome? In the paper, they set $\epsilon = \kappa \sqrt{d}$, while $\kappa$ is chosen by grid-search.

**Questions:**

Some minor questions:
1. How senstive of the W-CFM to the choice of cost function?
2. How senstive to $\epsilon$? According to simulation, looks quite sensitive?

---

> ### Author Response · Authors · 2025-11-19
>
> We thank the reviewer for the positive evaluation of the paper and for recognizing its main strengths. We address the remaining questions and concerns below. In the updated manuscript, new additions are marked in blue for convenience.
>
> ---
>
> > How sensitive of the W-CFM to the choice of cost function?
>
> ➡️ During our experimentation stage, we also evaluated our method using the squared Euclidean norm as the cost function and observed no statistically significant difference—the performance remains comparable to the Euclidean norm implementation on the same benchmarks.
>
> ---
>
> > How senstive to \eps? According to simulation, looks quite sensitive?
>
> ➡️ In order to get the best performance out of W-CFM, $\varepsilon$ must be chosen carefully. The heuristic we propose reliably guides the choice of an $\varepsilon$ that produces competitive results. To improve the robustness and transparency of our method, we have included the following in the revised version of the paper:
>
> * **a more general formulation of W-CFM, where the weight is now given by $w_\varepsilon (x,y) = \exp( - c(x,y) / \varepsilon) \hat f_\varepsilon (x) \hat g_{\varepsilon}(y)$**, with $\hat f_\varepsilon$ and $\hat g_\varepsilon$ being stochastic estimates of the tilting factors $f_\varepsilon, \hat g_\varepsilon$. The initial version of W-CFM that we introduced can be thought of as taking $\hat f_\varepsilon = \hat g_\varepsilon = 1$. In particular, we present a way to compute such estimates that is tailored to settings which feature strong tilting of the marginals (see Appendix C). As an example, we have updated our results on the 8 Gaussians →  moons problem (see the updated Table 1 and Table 5), which are significantly better than with the original method.
>
> * a transparent presentation of the algorithm for tuning $\varepsilon$ in **a new Appendix (Appendix D)**.
>
> * **an ablation study of \varepsilon on image generation for CIFAR10**, which we plan to extend to the rest of the datasets for the final version of the manuscript.
>
> ---
>
> ➡️ We thank the reviewer for the positive evaluation and constructive feedback. We hope that our clarifications and revisions satisfactorily address the remaining points.

---

### Official Review · Reviewer_vSxz · 2025-10-30

**Soundness:** 1
**Presentation:** 2
**Contribution:** 2
**Rating:** 2
**Confidence:** 4

**Summary:**

This paper proposes W-CFM, which leverages the Gibbs kernel as a weight function to reduce FM loss. The primary motivation for W-CFM is to straighten FM's path and thus improve FM's performance. The experimental results show that W-CFM performs well on unconditional generation in simulation and toy datasets such as CIFAR-10.

**Strengths:**

1. The topic of this paper is essential. Making the FM path straighter is key to reducing NFE and accelerating the sampling process while maintaining high generation quality. Reflow is one way, but it suffers from two-stage training and requires traversing the entire reverse process. W-CFM is one-stage and does not necessarily traverse the entire reverse process, which is interesting.

**Weaknesses:**

However, this paper contains some major concerns:

1. The motivation for introducing the Gibbs kernel to achieve a straighter path is unconvincing. Firstly, this paper does not offer any theoretical justification for this motivation. Meanwhile, the experimental results cannot support the motivation either. For example, Fig.2 shows that the path of OT-CFM is straighter than that of W-CFM. Meanwhile, image generation tasks also show that W-CFM cannot achieve this motivation. Under the same NFE, if the path is straight enough, the FID should show a "lerp". However, Fig. 3 indicates that the FID of W-CFM is similar to that of OT-CFM.

2. The motivation for using the Gibbs kernel itself is unclear. The related content spans 155-161 lines. But the author just claims that I-CFM can be written as a weight-function formulation. Then, they use the Gibbs kernel as another invariant of the weight function. Is there any specific reason here? In my view, directly leveraging a $l_{2}$ norm between GT and $x$ as the weight function can still work.

3. W-CFM is sensitive to the hyperparameter. As shown in Fig. 2, the hyperparameter directly determines W-CFM's performance, hindering its implementation in i2v and t2v tasks.

**Questions:**

1. Why cannot such a straight line accelerate the sampling process? The motivation for this question is that Rectified flow [1] mentions that a straighter path can reduce the NFE, which is the essence of Reflow. In this paper, a straighter path performs similarly to FM under the same NFE, according to Table 3, which is confusing.

2. Can the author report the performance of W-CFM in the hyperparameter=1 situation? The motivation for this question is that Fig. 2 shows that the smaller the hyperparameter, the straighter the path. A natural question is: what about using 1 directly?

3. Can you offer the experiments on different solvers, such as the Euler solver? The motivation for this question is that the Euler solver is the mainstream for FM, so we need to ensure W-CFG works well with it.

[1] Flow Straight and Fast: Learning to Generate and Transfer Data with Rectified Flow. Liu et al. ICLR 2022.

To sum up, the topic of this paper deserves more attention. But because of the weaknesses, it is difficult for me to be convinced that W-CFM can work. Therefore, I rate it as reject temporarily. If the author can clarify these concerns, I am willing to increase my score.

---

> ### Author Response · Authors · 2025-11-19
>
> We thank the reviewer for their thoughtful comments. In addressing the concerns raised, we have revised the manuscript and uploaded an updated version. For convenience, all significant changes are marked in blue.
>
> ---
>
> > The motivation for introducing the Gibbs kernel to achieve a straighter path is unconvincing. [...]
>
> ➡️ We thank the reviewer for an important remark that should be addressed carefully. We’d like to clarify the motivation for straighter paths.
> From a theoretical perspective, we omitted to mention **the link between entropic OT and Schrödinger bridge**, which is the main theoretical underpinning for obtaining low-energy trajectories via EOT and the reason why the Gibbs kernel naturally appears in our formulation. We have added the following paragraph to our revised version in section 2.1:
>
> *Entropic optimal transport is the static counterpart of the dynamic Schrödinger bridge problem [1]. The corresponding bridge defines a stochastic process whose drift (i.e., deterministic part) minimizes the energy along the path [2]. In other words, the Gibbs kernel appears naturally in EOT, which is a problem whose dynamic counterpart naturally implies minimizing the energy along trajectories, which incentivizes short and straight paths.*
>
> [1] Christian Leonard. A survey of the Schrodinger problem and some of its connections with optimal transport. arXiv preprint arXiv:1308.0215, 2013
> [2]  Nikita Gushchin, Alexander Kolesov, Alexander Korotin, Dmitry P Vetrov, and Evgeny Burnaev. Entropic neural optimal transport via diffusion processes. Advances in Neural Information Processing Systems, 36:75517–75544, 2023
>
> Secondly, we’d like to emphasize that we do not claim that our method achieves straighter paths than OT-CFM consistently; we claim that we are able to achieve straighter and shorter paths than I-CFM at virtually no cost. That is, without having to perform the mini-batch OT computations necessary in OT-CFM. To bring further context to Figure 2 (which has been updated), **we have also added the training times of the methods in Table 1**, and we observe that W-CFM is much faster to train than OT-CFM. Moreover, our method does not depend on batch size, so in some scenarios, it will perform better than OT-CFM. This can be seen in Figure 1: using too small a batch size for OT-CFM can lead to worse sample paths than what we achieve (also seen in Tables 2 and 3), further justifying the relevance of W-CFM.
>
> ---
>
> > The motivation for using the Gibbs kernel itself is unclear. [...]
>
> ➡️ We believe the motivation provided by entropic optimal transport, and in particular, Theorem 1, is enough to justify considering the Gibbs kernel weighting function. As highlighted at the beginning of section 3, introducing a weight is equivalent to computing the standard CFM loss with a prior distribution on $(X,Y)$ whose density is given by the weighting function. We then make the connection with Equation 2 (corresponding to the density of the EOT plan, which involves the Gibbs kernel) to justify using the Gibbs kernel as the weighting function. We want to highlight that the choice of the cost function is what eventually determines the exact form of the weighting function. In our paper, we focus on choosing $c(x,y) = || x - y ||$, giving the weight $w(x,y) = \exp (-|| x - y || / \varepsilon)$.
>
> ---
>
> > W-CFM is sensitive to the hyperparameter. [...]
>
> ➡️ Our method does require choosing $\varepsilon$ sensibly.
>
> However, to address the reviewer’s concern, we have included **a more general form of W-CFM: we allow arbitrary choices of tilting factors $\hat f_\varepsilon$ and $\hat g_\varepsilon$ in the weighting function, i.e., $w_\varepsilon(x,y)$ $= \exp (-c (x,y) / \varepsilon)$ $\hat f_\varepsilon (x)$ $\hat g_\varepsilon (y)$**. This allows us to limit the effect of the bias for small values of $\varepsilon$, effectively making our method more robust to different choices of $\varepsilon$. Details can be found in **the updated version of Section 3.1** in the revised PDF.
>
> Moreover, we have included **an explicit algorithm for choosing $\varepsilon$ in Appendix D**, which hopefully makes our method more transparent and applicable to more general cases. We have also included **an ablation study for $\varepsilon$ on image generation in Appendix E**, which shows that one can still get decent results even with poor tuning of $\varepsilon$.
>
> ---

---

> ### Author Response · Authors · 2025-11-19
> **Rebuttal Part ii**
>
> > Why cannot such a straight line accelerate the sampling process? [...]
>
> ➡️ This point is essential and requires clarification. In low-dimensional settings, the effect of techniques such as OT-CFM, ReFlow and W-CFM is clear: the paths get straighter because during training, the linear interpolation paths between the sample pairs do not cross as much as when using I-CFM. However, in high-dimensional settings, **these paths do not cross** even when using I-CFM, especially in the region away from the source and target distributions. Hence, the straightening effect is only noticeable near the source and target distributions. This can be seen, for instance, in Figure 6 of [1]. More importantly, in these high-dimensional settings, OT-CFM and W-CFM tend to reduce the overall path length, which means that the errors are less likely to accumulate for a similar level of discretization than when using I-CFM.
>
> We want to highlight that, across our experiments, in both toy and image settings,  **W-CFM is almost always superior to I-CFM across different integration schemes (Euler and Dopri5)**.
>
> [1] Flow Straight and Fast: Learning to Generate and Transfer Data with Rectified Flow. Liu et al. ICLR 2022.
>
> ---
>
> > Can the author report the performance of W-CFM in the hyperparameter=1 situation? [...]
>
> ➡️ The choice of the values of $\varepsilon$ depends on the scale of the typical distances between source and target distributions. We decided to include results for the 8 Gaussians ->  moons task for $\varepsilon$ ranging in {2, 4, 6, 8, 10}, as we believe this range best illustrates the tilt–straightness tradeoff. We emphasize that **we have updated the results for this experiment (Table 1, Figure 2, Table 5, and Figure 4)**, and that all values of $\varepsilon$ now benefit from improved sample quality due to the simple numerical scheme we introduce to mitigate marginal tilting. This scheme is a special case of **the more general W-CFM presented in Section 3.1** and is described explicitly in **Appendix C**.
>
> Regarding the question about using $\varepsilon = 1$: for this particular experiment, choosing such a small $\varepsilon$ would typically induce strong marginal tilting and consequently poor sample quality, which is precisely why we rely on our heuristic for selecting a sensible value of $\varepsilon$ (see Section 3.2.1 and **the newly added Appendix D** to understand our heuristic further). We now make this clearer in the revised text. As an additional illustrative example, **we have also added an ablation on CIFAR-10 in Appendix E** that includes the case $\varepsilon = 1$.
>
> ---
>
> > Can you offer the experiments on different solvers, such as the Euler solver?
>
> ➡️ All the results in Table 3 are given using the Euler solver with different NFEs. As can be seen in this table, W-CFM outperforms the other methods on most datasets and for most values of NFE.
>
> ---
>
> ➡️ As you indicated that you were open to increasing the score pending clarification, we hope the revisions and added experiments resolve the points you raise, and we would appreciate a reconsideration of the score.

---

> > ### Comment · Reviewer_vSxz · 2025-11-28
> >
> > Thanks to the author`s rebuttal. I have carefully checked all the content, including the revised manuscript (This takes some time). I acknowledge that the improved manuscript addresses many concerns, including the motivation for introducing the Gibbs kernel and why it should be used. But after further clarification, the concerns persist in practice. Concretely, 1) Even though the W-CFM is better than the I-CFM in theory, it still cannot improve the image generation quality. The evidence is that the improvement of FID made by W-CFM is marginal compared to that made by I-CFM, as shown in Table 3. 2) Table 1 is to supports that W-CFM is batch-size agnostic and thus could use the small batch size to accelerate the training process, which is the essential evidence to support that W-CFM is better than I-CFM. However, enabling a small batch size does not necessarily accelerate convergence during training. In this case, Table 1 cannot support the claim that W-CFM is better than I-CFM in practice.
> >
> > To sum up, I acknowledge that the proposed W-CFM makes a theoretical contribution. But there is a large gap between the theory and practice. Therefore, this paper falls short of the acceptable bar, and I have to keep my initial score.

---

### Official Review · Reviewer_Ttjj · 2025-10-30

**Soundness:** 2
**Presentation:** 2
**Contribution:** 2
**Rating:** 2
**Confidence:** 3

**Summary:**

The paper introduces Weighted Conditional Flow Matching (W-CFM). The key idea is to weight each training pair using a Gibbs kernel, allowing the method to approximate the Entropic Optimal Transport (EOT) plan without the high computational cost of explicitly solving the OT problem and without being constrained by batch size. The authors derive theoretical conditions under which this approximation preserves the original marginals and prove that W-CFM becomes equivalent to OT-CFM in the large-batch limit, provided the marginals remain unchanged. The method is evaluated on both toy and image datasets.

**Strengths:**

The authors propose a method that approximates EOT-CFM by simply reweighting pairs sampled from an independent plan, thus avoiding the need for mini-batch OT computations. They provide a theoretical justification for its equivalence to OT-CFM in the large-batch limit. However, this reweighting results in tilted marginals, i.e., $\tilde{\mu}_\epsilon(dx)=\frac{\exp{(-\phi\_\epsilon(x))}}{Z^1\_\varepsilon}\mu(dx)$ (eq. 10), where $\mu(dx)$ is the original marginal, so the authors analyze the conditions under which this tilting can be neglected. They also introduce a procedure for selecting the regularization coefficient $\varepsilon$, which controls the trade-off between the straightness of trajectories and the degree of marginal tilting in the proposed framework. Extensive experiments show that the proposed method achieves performance comparable to previous approaches.

**Weaknesses:**

### **Terminology**

- In the statement of Theorem 1, there are $\phi, \psi$, whereas in Equation (2) they appear as $\phi_\varepsilon, \psi_\varepsilon$. Moreover, Theorem 4.2 from [1] states that $\phi, \psi$ are called _EOT potentials_, not _Schrödinger potentials_ and Remark 4.3 therein notes that there is already some inconsistency in the naming.

### **Theory**
- As I understand it, $\mathcal{L}\_{\text{W-CFM}}$ can be considered an accurate approximation of $\mathcal{L}\_{\text{EOT-CFM}}$ only if the EOT potentials corresponding to the entropic plan satisfy

$$-\phi\_\varepsilon(x)\psi_\varepsilon(y) \approx C$$

which seems a very restrictive assumption meaning that the potentials are constants.

- Furthermore, there is a notation overload in Equation (12), since $f_\varepsilon(x)$ was already defined as $f_\varepsilon = \exp(\phi_\varepsilon(x))$. The conditions below (12) for marginal preservation are rather broad and obvious. It is not clear how the constant between the two measures affects the tilting – it seems that this constant should be close to 1, but the sensitivity of this constant is not discussed.

- Proposition 3 appears rather trivial, as the assumption that the tilted marginals are not tilted is too strong. In this case, $q_\varepsilon = \pi_\varepsilon$, since $\exp(-\phi_\varepsilon(x)) = Z^1_\varepsilon$ and $\exp(-\psi_\varepsilon(y)) = Z^2_\varepsilon$, and therefore, from Equation (8), it follows that $Z_\varepsilon = Z^1_\varepsilon Z^2_\varepsilon$. For the case $q_\varepsilon = \pi_\varepsilon$, the mini-batch approximation of the true entropic plan is a standard result [2]. Moreover, the proof relies mainly on intuitive reasoning and does not provide references for rigorous derivations.

---

### **Practice**

- W-CFM has a notable limitation – a trade-off controlled by $\varepsilon$ between preserving the marginals and maintaining straight trajectories. In Table 5, larger values of $\varepsilon$ prevent marginal distortion but result in less straight trajectories. Since $\varepsilon$ appears under the exponent, the tilting of marginals seems to be very sensitive to its value. Moreover, I did not find a sensitivity analysis for $\varepsilon$ on the image domain.

- Finally, the performance improvement is modest, as shown in Table 2, where even I-CFM achieves comparable results.

[1] Nutz, Marcel. "Introduction to entropic optimal transport." Lecture notes, Columbia University (2021).

[2] Hundrieser, Shayan, Marcel Klatt, and Axel Munk. "Limit distributions and sensitivity analysis for empirical entropic optimal transport on countable spaces." The Annals of Applied Probability 34.1B (2024): 1403-1468.

**Questions:**

- In Table 2, the performance of the proposed method is compared with I-CFM and OT-CFM. Could the authors also report the training time to clarify how large is the computational advantage of the proposed method?
- It would also be helpful if the authors included results with different cost functions, such as the $\ell_1$ distance, though this is minor.
- In addition, could the authors provide an analysis of the trade-off with respect to $\varepsilon$ on image data?
- Finally, how is this work related to [1] (Theorem 4.3) and [2] (Eq. 10)?

[1] Zhang, Shiyuan, Weitong Zhang, and Quanquan Gu. "Energy-Weighted Flow Matching for Offline Reinforcement Learning." ICLR 2025.
[2] Park, Seohong, Qiyang Li, and Sergey Levine. "Flow q-learning."  ICML 2025.

---

> ### Author Response · Authors · 2025-11-19
>
> We appreciate the reviewer’s insightful feedback. In addressing the concerns raised, we have revised the manuscript and uploaded an updated version. For convenience, all significant changes are marked in blue.
>
> ---
>
> > In the statement of Theorem 1, there are \phi \psi [...]
>
> ➡️ We thank the reviewer for pointing out the inconsistencies; we have made sure they are addressed in the revised version.
>
> ---
>
> > As I understand it, L_W-CFM can be considered an accurate approximation [...]
>
> ➡️ From a theoretical perspective, we definitely agree that constant potentials / no tilting is a strong assumption.
>
> As a solution to this theoretical limitation, **we have generalized our W-CFM loss to allow for the incorporation of stochastic estimates of the tilting factors $f_{\varepsilon}$ and $g_{\varepsilon}$** directly inside the weight, i.e. $w_\varepsilon(x,y) = \exp (-c (x,y) / \varepsilon)$ $\hat f_\varepsilon (x) \hat g_\varepsilon (y)$ (details can be found in Section 3.1 of the revised pdf).
>
> We have consequently introduced a **new Proposition (Proposition 1)** to formalize this general approach. In particular, **we present one method of computing such estimates via Monte Carlo in Appendix C**, and we are pleased to report **improved results on the 8 Gaussians -> Moons task (see Table 1 and Table 5)**.
>
> However, we argue that many practical high-dimensional settings are favourable to our original method. On one hand, the concentration of measure for high-dimensional Gaussians suggests that typical high-dimensional normalized distributions have most of their support on a sphere, which corresponds to the setting of Proposition 3 (formerly Proposition 2). On the other hand, our experiments on image generation suggest that we can still pick \eps to be small with respect to the dimensionality of the data, and still get good performance. This means that the marginal tilting is at worst a benign phenomenon that does not affect the performance of the model (provided one chooses \eps carefully) as shown in Tables 2, 3, and 4.
>
> ---
>
> > Furthermore, there is a notation overload in Equation (12) [...]
>
> ➡️ The densities of the tilted marginals with respect to the original ones are indeed crucial, as they correspond to an a priori measure of the quality of the samples for a model trained with W-CFM. We want to make it clear that we do use these densities to choose the value of $\varepsilon$ adequately. **We have included a detailed algorithm for choosing $\varepsilon$ based on MC estimates of these densities in Appendix D.**
>
> ---
>
> > Proposition 3 appears rather trivial [...]
>
> ➡️ We acknowledge the fact that Proposition 4 (formerly Proposition 3) is an expected result. It essentially serves as a way to rigorously connect our W-CFM method in an ideal setting with the existing OT-CFM with mini-batch EOT coupling. As for the rigour of proof, we believe the proof given in Appendix A.4 (formerly Appendix A.3) is of an acceptable standard and does provide adequate references when needed. We want to highlight that Appendix B, which also deals with the connection to OT-CFM, only provides an informal discussion for the reader who wants to get a high-level overview of the connection, and does not constitute a proof of Proposition 4.
>
> ---
>
> > W-CFM has a notable limitation [...]
>
> ➡️ The sensitivity of our method to the choice of \varepsilon is indeed crucial and should be handled carefully, as we discuss in Section 3.2.1. In the revised version of the paper, we make our heuristic for choosing \eps more explicit by **adding an algorithm in Appendix D detailing the relative variance criterion**. Moreover, the new formulation of W-CFM with the stochastic estimates of the tilting factors makes our method more robust to cases where tilting might be significant, as shown in our **updated results on 8 Gaussians → moons (Table 1 and Table 5)**.
>
> Finally, **we run an ablation study of $\varepsilon$ in Appendix E in the CIFAR-10 experiment**, which we plan to extend to the other datasets for the final version of the manuscript.
>
> ---
>
> > Finally, the performance improvement is modest [...]
>
> ➡️ While we acknowledge that the performance improvement might not look impressive, we want to highlight the fact that on all datasets considered, for almost all configurations (choice of solver and NFE), W-CFM outperforms I-CFM, while requiring little to no overhead computations. We believe these results support our overall claim that W-CFM leads to better quality flow models than I-CFM at no cost.
>
> ---
>
> > Could the authors also report the training time to clarify how large is the computational advantage of the proposed method?
>
> ➡️ We agree with the relevance of adding training time to our reported results. We have added them to the toy datasets, **see Table 1**, where the speedup vs. OT-CFM is significant.
>
> ---

---

> ### Author Response · Authors · 2025-11-19
> **Rebuttal part ii**
>
> ---
>
> > It would also be helpful if the authors included results with different cost functions, such as the distance, though this is minor.
>
> ➡️ As indicated by the general formulation of our method, W-CFM supports a range of cost functions. For the problems we study, the Euclidean norm seems like the most natural choice. We tried the method using the squared Euclidean norm as the cost function and observe no statistically significant difference—the performance remains comparable to the Euclidean norm implementation on the same benchmarks.
>
> ---
>
> > In addition, could the authors provide an analysis of the trade-off with respect to $\varepsilon$ on image data?
>
> ➡️ We have added **an ablation study of $\varepsilon$ on unconditional CIFAR10 image generation in Appendix E**, where the effect of $\varepsilon$ on FID corresponds to what is expected. We plan to run this ablation for the remaining datasets for the final version of the manuscript. We hope this brings transparency to our method.
>
> ---
>
> > Finally, how is this work related to [1] (Theorem 4.3) and [2] (Eq. 10)?
>
> ➡️ We thank the reviewer for pointing out relevant references. Theorem 4.3 in [1] is indeed quite similar to our W-CFM, and the authors also make a connection to importance sampling. Here are the main differences between our approaches:
> In [1], the authors are motivated by learning a model that directly samples from a tilted target distribution, where the tilting is known beforehand, whereas we try to generate straighter paths for a base model by relying on the a priori unknown EOT plan.
> Moreover, the energy functional considered in [1] is only applied to the target sample and has little to no effect on the geometry of the path. In our case, we are introducing a weighting scheme that takes as input both endpoints, and leads to a straightening of the paths (and contrary to [1], we try to avoid marginal tilting).
>
> Equation 10 in [2] is similar to the example presented in Section 5 in [1]. In that case, the goal is to learn policies using flow or diffusion models. During training, one wants the learned policy to generate actions that yield high rewards, hence one biases the behavioural cloning loss towards transitions (s, a) that generate high rewards (in [1] they use the Q-function, in [2] they use the advantage function). The authors in [2] go one step further by essentially learning a consistency model (see [3]) on top of a base flow policy, which is also a way of speeding up inference that we acknowledge in the related works section.
>
> In the revised version, **we make sure to mention the connections between our work and these two papers in a new paragraph of the Related Work section**.
>
> [1]  Zhang, Shiyuan, Weitong Zhang, and Quanquan Gu. "Energy-Weighted Flow Matching for Offline Reinforcement Learning." ICLR 2025.
> [2] Park, Seohong, Qiyang Li, and Sergey Levine. "Flow q-learning." ICML 2025.
> [3] Song, Yang, Prafulla Dhariwal, Mark Chen, and Ilya Sutskever. "Consistency models." ICML 2023.
>
> ---
>
> ➡️ We have made substantial revisions to address the points you raised, including clarifying the theory, improving the exposition, and adding new experiments and ablations. We hope these changes reflect the merit of the work more accurately, and we would appreciate a reconsideration of the score.

---

> > ### Comment · Reviewer_Ttjj · 2025-11-27
> > **Response**
> >
> > Dear Authors,
> >
> > Thank you for your detailed response. After reading them, I gave one more read for the paper (its updated version). While some of my concerns have been addressed, several issues remain and even new questions appeared.
> >
> > - **Experimental demonstrations:** The authors state: “We believe these results support our overall claim that W-CFM leads to better quality flow models than I-CFM at no cost.” However, I think that this claim is not sufficiently supported for a primarily empirical paper, especially given its known theoretical issues like bias. For instance, the reported CIFAR-10 FID of 7.33 for W-CFM after 400k iterations is not even competitive with the FID of ≈3.5 achieved by OT-CFM [1] after 500k iterations. While the authors state their "goal is not to reach state-of-the-art performance," a comparison should still approach well-established settings (FID level) for the model type (e.g., [1]). Furthermore, the lack of code in the supplementary significantly undermines the study's reliability and reproducibility. Overall, the analysis could have been much stronger if the authors provided convergence plots in FID and its final std FID (for different training runs of the model) at least for some image-based experiments. With all these gaps, I still believe the results are not convincing enough.
> >
> > - **Theoretical presentation:** The theoretical results are not fully convincing and appear inaccurate in my view. For instance, in Eq. (8) of the revised version, the authors use $\frac{\mathbb{E}\hat{f}(x)}{f(x)}$. If $\hat{f}(x)$ is an unbiased estimator of $f(x) (\mathbb{E}\hat{f}(x) = f(x))$, this ratio is equal to $1$. Additionally, it is unclear which marginals the EOT plan $\pi_\varepsilon$ is defined between, leaving the reader to infer this. Such statements are really vague and confusing.
> >
> > - **Minor experimental comments:**
> >
> >     1. **Table 1 (8 Gaussians $\to$ Moons):** It is not clear, whether the reported time includes the precomputation of dual potentials.
> >
> >     2. **Table 5:** Computation time is not reported. The table starts with $\varepsilon = 2$. I wonder if the results for smaller $\varepsilon$ could be provided. For high $\varepsilon$, the optimal transport plan seems very stochastic and similar to independent, so this experiment may be not representative.
> >
> >     3. **Image datasets:** Experiments with dual potential precomputation are missing.
> >
> > **Summary:**
> >
> > While the paper proposes an interesting idea, its practical implementation raises significant concerns. Some theoretical explanations are trivial or confusing. I believe the paper requires major revision and a fresh round of review and is not yet ready for publication. I have therefore kept my score unchanged.
> >
> > **P.S.** I also recommend that the authors consider method [2], as used in [3], to learn the potentials as an alternative to the Monte Carlo estimations proposed in the paper.
> >
> > ---
> > **References**
> >
> > [1] Tong, Alexander, et al. "Improving and generalizing flow-based generative models with minibatch optimal transport." _Transactions on Machine Learning Research_ (2024): 1-34.
> >
> > [2] Seguy, Vivien, et al. "Large-Scale Optimal Transport and Mapping Estimation." _ICLR 2018-International Conference on Learning Representations_. 2018.
> >
> > [3] Daniels, Mara, Tyler Maunu, and Paul Hand. "Score-based generative neural networks for large-scale optimal transport." _Advances in neural information processing systems_ 34 (2021): 12955-12965.

---

### Official Review · Reviewer_st92 · 2025-10-31

**Soundness:** 2
**Presentation:** 3
**Contribution:** 1
**Rating:** 2
**Confidence:** 4

**Summary:**

The authors propose the method for learning the Conditional Flow Matching models with straighter and shorter trajectories. The authors propose to modify the optimization objective by incorporating the Gibbs kernel, i.e., weightening of training pairs (x, y), and call their method Weighted Conditional Flow Matching (W-CFM). The modified objective coincides with the objective of Conditional Flow Matching with Entropic Optimal Transport plan for training pairs (x, y), but with different “tilted” marginals. Authors address the issue with “tilted” marginals in practice by optimizing over the stochasticity of EOT parameter $\epsilon$ and show examples where marginals are not “tilted”. In addition, the authors show that under some assumptions their method W-CFM coincides with the EOT-CFM method in the limiting batch size case.

On practice authors, compare their method with other Conditional Flow Matching methods with different data couplings. The methods are compared on toy 2D datasets as well as on high dimensional image datasets.

**Strengths:**

- The authors propose a method for training a CFM model that lets one to approximate an entropic optimal transport plan for training data pairs, without extra compute as in OT-CFM/EOT-CFM.

- The authors test their approach on several data types and analyze different aspects of CFM models.

- The method delivers decent results on image data, outperforming other CFM approaches by both fidelity and diversity.

**Weaknesses:**

The main weakness of the method is **“tilted” marginals**:

- There are no clear types of cases where marginals are not tilted (except for very limited case described in Proposition 2). In contrast the OT-CFM/EOT-CFM methods are asymptotically unbiased and when batch size grows to infinity they recover the ground truth OT or EOT plan. While in W-CFM there are no such parameters that guarantee the lack of tilting.

- In that light the claims of authors that distributions are not “tilted” significantly remain unsubstantiated.

- Proposition 3 is built on a “mild regularity” assumption (no marginal tilting and bounded support) and in my opinion these assumptions are not “mild” at all, since authors show very few cases where marginals are not “tilted” and bounded support is also a strong assumption.

The practical methodology for choice of $\epsilon$ **remains unclear.** Authors say “we search over a grid of k values spaced uniformly in log scale and select the smallest value for which the relative variance starts flattening, following an ”elbow rule” heuristic akin to the selection of the number of principal components in PCA”. But still I do not catch the overall algorithm. Can authors explain in more details?:
- As far as I understand the bigger the $\epsilon$ the closer ratios between “tilted” and original marginals, i.e, $f_\epsilon$ and $g_\epsilon$ in Eq 12, to constant. In that sense the bigger $\epsilon$ one takes the less “tilted” the marginals, but this undermines W-CFM purpose, because then the EOT plan is closer to an independent plan. Then the choice of $\epsilon$ remains even more controversial and unclear.

The experiment results are **ambiguous.**
- On the MoG -> 5 Gaussians problem W-CFM delivers not the straightest trajectories (Table 1, NPE), but shows the best distributions fit (Table 1, W2)
- While on 8 gaussians -> moons the W-CFM is worse in both fitting the target distribution and trajectories straightness.
- While on image dataset the W-CFM in majority cases indeed deliver better fidelity and diversity then I-CFM and OT-CFM, it holds not in all the cases and the gains w.r.t. other methods are rather marginal.
- The authors compare their method with OT-CFM, but I couldn't find information wherever they use mini batch Optimal Transport or mini batch Entropic Optimal Transport (Sinkhorn) to generate training data pairs. While the comparison with mini batch EOT with the same $\epsilon$  would be desirable for clean experimental comparison.

**Questions:**

- Can authors explain the methodology for choosing $\epsilon$ more thoroughly and extrapolate on lines 253-255?

- Which mini batch pairing algorithm, i.e., mini batch Optimal Transport or mini batch Entropic Optimal Transport with some $\epsilon$, was used as a part of OT-CFM in the experimental part of the paper?

---

> ### Author Response · Authors · 2025-11-19
>
> We thank the reviewer for their thoughtful comments. In response to the reviewers’ concerns, we have prepared and uploaded an updated version of the manuscript. For ease of reference, all notable revisions are highlighted in blue.
>
> ---
> > The main weakness of the method is “tilted” marginals
>
> ➡️ It is true that having perfect marginal recovery (i.e., no tilting) only happens in limited cases a priori.
>
> In the revised submission, **we extend W-CFM to allow arbitrary choices of tilting factors $\hat f_\varepsilon$ and $\hat g_\varepsilon$ in the weighting function, i.e., $w_\varepsilon(x,y)$ $= \exp (-c (x,y) / \varepsilon)$ $\hat f_\varepsilon (x)$ $\hat g_\varepsilon (y)$** (details can be found in Section 3.1 of the revised pdf).
>
> We present these as stochastic estimates of the tilting factors for the source and target distributions. We go over one simple method for incorporating such estimates using Monte-Carlo simulation, in a **new section of the Appendix (Appendix C)**. This method is relevant in the examples which feature important tilting, e.g., the 8 Gaussians -> moons task. As reported in **the updated Tables 1 and 5**, incorporating these estimates produces substantial improvements in sample quality, confirming that we can correct marginal tilting effectively.
>
> **While the above correction is important when tilting is non-negligible, this behaviour does not carry over to our high-dimensional settings.** Here, for a good choice of $\varepsilon$, we observe that marginal tilting is extremely mild, and that the simple choice $\hat f_\varepsilon = \hat g_\varepsilon = 1$ is practical for two reasons:
>
> i) **Empirical evidence:** Across all image generation benchmarks, W-CFM achieves FID scores comparable to OT-CFM, consistently outperforms I-CFM, and does this with no degradation in the diversity metrics (see Table 5). If marginal tilting were significant, these metrics would deteriorate, yet they do not.
>
> ii) **High-dimensional geometric intuition:** As we discuss in the paper, in high dimension, Gaussian/normalized data concentrates near a thin shell leading to almost constant potentials. Also, by choosing $\varepsilon$ with our heuristic, we ensure that the tilting factors are approximately constant (their relative variance is on the order of $10^{-2}$), under these conditions, the marginal tilting is minimal as we empirically observe.
>
> We acknowledge that this phenomenon occurring in high-dimensions is only alluded to after Proposition 2 and should be stated clearly in the experiments section. We make sure to highlight this important observation in the revised version of the paper (see last paragraph of 5.3).
>
> ---
>
> > Proposition 3 is built on a "mild regularity" assumption...
>
> ➡️ We acknowledge that the wording might not be adequate in Proposition 4 (formerly Proposition 3), we made sure to clearly indicate that this result is just a way of establishing a rigorous connection to an existing CFM method in an ideal theoretical setting.
>
> ---
>
> > The practical methodology for choice of \eps remains unclear. [...]
>
> ➡️ The choice of $\varepsilon$ is indeed crucial in our method, and you are right to highlight the tradeoff between marginal tilting and straightness of paths. We argue that it is possible to find good values of $\varepsilon$, where one gets straighter trajectories than I-CFM while retaining good sample quality. First, we argue that the choice of \eps should be guided by the typical value of the cost: in our case, with the Euclidean cost function, the typical value is $\sqrt{d}$. We then propose a heuristic method to refine the choice of $\varepsilon$ and look at values of the form $\varepsilon = \kappa\sqrt{d}$: we compute a Monte Carlo approximation of the relative variance of the tilting factors $\tau_{\mu, \varepsilon}$ and $\tau_{\nu, \varepsilon}$. If these tilting factors were constant, then there would be no marginal tilting; therefore, choosing $\varepsilon$ such that their relative variance is close to zero is a natural criterion. In practice, we sweep $\kappa$ over a logarithmic grid and select the smallest value for which the relative variance curves flatten (an “elbow rule”).
>
> **In the revised version of the paper, we include an appendix (Appendix D) with the detailed algorithm in practice, in the hopes of making our method more transparent and reproducible.**
>
> ---

---

> ### Author Response · Authors · 2025-11-19
> **Reply part ii**
>
> > The experiment results are ambiguous. [...]
>
> ➡️ As discussed above, in the revised version of the paper, we introduce a scheme to correct the tilting, which we apply on the 8 Gaussians -> Moons experiment and which noticeably improves the results. As shown in the updated Tables 1 and 5, using these estimated tilting factors allows W-CFM to outperform I-CFM on both sample quality and straightness, while remaining far cheaper to train than OT-CFM (we now report training times to highlight the cost difference). These examples clarify how sensitive this low-dimensional benchmark is and motivate the importance of choosing $\varepsilon$ appropriately.
>
> All in all, on the high-dimensional datasets, we retain strong performance in terms of both FID and diversity metrics, with minimal computational overhead during training (in particular, without computing mini-batch OT). Most notably, we consistently outperform I-CFM.
>
> As for a comparison with mini-batch EOT with the same values of $\varepsilon$, we believe that it would not depict the performance of OT-CFM faithfully, as the typical values used with the Sinkhorn algorithm are much smaller than the ones considered when working with image data (see for instance [1]).
>
> [1] Zhang, Stephen, Alireza Mousavi-Hosseini, Michal Klein, and Marco Cuturi. "On fitting flow models with large sinkhorn couplings." arXiv preprint arXiv:2506.05526 (2025).
>
> ---
>
> > The authors compare their method with OT-CFM, but I couldn't find information [...]
>
> ➡️ We acknowledge that we should have presented the OT-CFM results with greater clarity: the mini-batch pairing was done by solving the exact optimal transport problem. We have included this clarification in the revised submission.
>
> ---
>
> > Can authors explain the methodology for choosing \eps [...]
>
> ➡️ We hope we addressed this question in our discussion above and the **newly added Appendix D** in the revised manuscript.
>
> ---
>
> > Which mini batch pairing algorithm, i.e., mini batch Optimal Transport or mini batch Entropic Optimal Transport with some , was used as a part of OT-CFM in the experimental part of the paper?
>
> ➡️ We also hope we addressed this concern in our discussion above.
>
> ---
>
> ➡️ We hope the revisions (adding the theoretical clarifications, the detailed ε-selection algorithm, and the improved experimental results) address the concerns you raised. We would appreciate it if you could reconsider the score based on the updated manuscript (see all changes in blue).

---

> > ### Comment · Reviewer_st92 · 2025-11-27
> > **Thank you for the answer**
> >
> > I thank authors for taking the time to answer my concerns.
> >
> > The introduction of Monte Carlo approximation of tilting factors seem interesting and promising. However, it adds computational cost and then seems like the whole computation cost of the method the same as the FM with precomputed mini batch EOT, but nevertheless I think this is promising.
> >
> > In addition, regarding the tilted marginals, I agree that seems like tilting is negligible on image data. However, I think the main reason for that is high epsilon values. As you have mentioned yourselves, typical epsilon values used in the Sinkhorn algorithm are much smaller than the ones you suggest. I believe, such small values usually taken because they provide relatively straight paths, in contrast to higher epsilon values explored in this paper. Furthermore, I think that meaningful experiments with lower epsilon parameter are essential to provide meaningful comparison with OT-CFM and EOT-CFM.
> >
> > The other answers do mostly satisfy me, and it is nice to see them in the revision. I acknowledge that authors has strengthened their manuscript in the revised version.
> >
> > However, in general, I still think that the method is limited, because:
> > * marginals tilting bias in the solution
> > * large eps values to mitigate this bias
> > * marginal performance gains and as a consequence limited practical use
> >
> > In addition, the overall placement of the method is vague since it cannot be applied to unpaired domain translation, because of bias, and doesn't provide a lot of gains on the image data generation. Maybe some other set of experiments that more expressively highlights the W-CFM benefits w.r.t. OT-CFM and I-CFM would be more convincing.
> >
> > In that light, I am willing to keep my score.

---

### Author Response · Authors · 2025-11-20
**General Comment**

We thank all reviewers for their constructive feedback, which has motivated us to make **substantial improvements** and include **new results** that we believe further strengthen the paper’s overall contribution. Below, we summarize the strengths highlighted by each reviewer and outline the concrete additions and clarifications incorporated into the revised submission.

---

### **Main strengths highlighted by the reviewers**

- **Low computational cost compared to OT-CFM**. All reviewers noted that W-CFM avoids mini-batch OT and trains as cheaply as I-CFM.

- **Ability to approximate EOT-CFM without mini-batch OT**. Reviewers st92 and Ttij highlighted that W-CFM recovers the behavior of EOT-CFM without solving the OT problem.

- **Competitive or better empirical performance**. Reviewers st92 and MqhE acknowledged that W-CFM achieves competitive or superior results on image datasets in terms of fidelity and diversity.

- **Clarity and readability of the paper**. Reviewers Ttij and MqhE explicitly noted that the paper is clear and easy to follow.

---

### **Main additions and revisions in the updated manuscript (see changes in blue)**

- A generalized formulation of W-CFM allowing arbitrary tilting factors $\hat f_\varepsilon,\hat g_\varepsilon$ inside the Gibbs weight.

- A Monte Carlo estimation procedure for the tilting factors (Appendix C), applied to settings where tilt is non-negligible (e.g., 8 Gaussians → moons), with updated results in Tables 1 and 5.

- A new theoretical result (current Proposition 1) which clarifies that if $\hat f_\varepsilon$ and $\hat g_\varepsilon$ are unbiased (up to constants), the W-CFM objective is proportional to the standard CFM loss computed using the optimal EOT plan $\pi_\varepsilon$.
This formally connects our generalized weighting scheme to existing CFM theory.

- A concrete algorithm for choosing $\varepsilon$ via relative-variance diagnostics (Appendix D), addressing requests for a clearer tuning methodology.

- Updated results on 8 Gaussians → moons using the new MC estimators and additional results for the training times of each of the methods (Tables 1 & 5; Figures 2 & 4).

- An ablation of $\varepsilon$ on CIFAR-10 (Appendix E).

- Several notational and terminology corrections based on reviewer feedback.

---

We hope that the revisions and clarifications above address the concerns raised in the reviews. We are happy to answer any remaining questions during the discussion phase, and we would appreciate it if the reviewers could reconsider their scores in light of the updated manuscript.

---

### Meta-Review · Area_Chair_y6qt · 2025-12-10

**Summary:**

Based on entropic optimal transport, this paper proposes a Weighted Conditional Flow Matching that modifies the classical CFM loss by weighting each training pair with a Gibbs kernel. This paper establishs an equivalence between W-CFM and the minibatch OT method in the large-batch limit, showing how their proposed method overcomes computational and performance bottlenecks linked to batch size.

**Reviewer Concerns:**

The main weakness of the method is “tilted” marginals.

The practical methodology for choice of $epsilon$ remains unclear.

 The authors state: “We believe these results support our overall claim that W-CFM leads to better quality flow models than I-CFM at no cost.” However, reviewer think that this claim is not sufficiently supported for a primarily empirical paper, especially given its known theoretical issues like bias.

The theoretical results are not fully convincing and appear inaccurate

**Reviewer Scores:**

Most reviewers keep their scores.

---

### Decision · Program_Chairs · 2026-01-26

Reject